# MolPhase, an advanced prediction algorithm for protein phase separation

Qiyu Liang [ID] [1,2,4], Nana Peng [ID] [2,4], Yi Xie[2], Nivedita Kumar[2], Weibo Gao[1] & Yansong Miao [ID] [2,3][✉]

## Abstract

**We introduce MolPhase, an advanced algorithm for predicting protein phase separation (PS) behavior that improves accuracy and reliability by utilizing diverse physicochemical features and extensive experimental datasets. MolPhase applies a user-friendly interface to compare distinct biophysical features side-by-side along protein sequences. By additional comparison with structural predictions, MolPhase enables efficient predictions of new phase-separating proteins and guides hypothesis generation and experimental design. Key contributing factors underlying MolPhase include electrostatic pi-interactions, disorder, and prion-like domains. As an example, MolPhase finds that phytobacterial type III effectors (T3Es) are highly prone to homotypic PS, which was experimentally validated in vitro biochemically and in vivo in plants, mimicking their injection and accumulation in the host during microbial infection. The physicochemical characteristics of T3Es dictate their patterns of association for multivalent interactions, influencing the material properties of phase-separating droplets based on the surrounding microenvironment in vivo or in vitro. Robust integration of MolPhase's effective prediction and experimental validation exhibit the potential to evaluate and explore how biomolecule PS functions in biological systems.**

**Keywords** Phase Separation; Molecular Condensation; Prediction; Effector
**Subject Categories** Computational Biology; Methods & Resources; Plant Biology

## Introduction

Protein phase separation (PS) in biological systems is a complex and spatiotemporally regulated process to create functional membraneless organelles. The initiation and propagation of homotypic PS are determined by the intrinsic physicochemical properties of each biomolecule, which will also be significantly influenced by the abundance, localization, and components in living systems. When multiple macromolecular components condense, the resulting heterotypic phase separation is influenced by the complex interactions of each component. This is determined by balancing their individual biophysical characteristics and associative behaviors (Pappu et al, 2023). To effectively analyze the propensity for protein phase separation based on intrinsic physicochemical features is crucial. This method aids in formulating informed hypotheses by also considering their in vivo macromolecular environments. The initiation of protein cluster formation under subsaturation conditions is a critical transition step before evolving to high-order assembly and then the near-equilibrium phase separation, which is often shown by in vitro biochemical assays (Lan et al, 2023; Lee et al, 2023; Seim et al, 2022). The formation of initial molecular assemblies and their evolution into condensates are largely influenced by the intrinsic biophysical properties of scaffolding proteins. These processes start at a nanometer scale, making them difficult to quantify. This challenge is further compounded by the need for high spatiotemporal resolution in microscopy and sensitivity in single molecule detection. An effective prediction of homotypic PS using biophysical features of protein sequences has shown great potential to guide hypothesis generation and experimental design. Combining existing knowledge of PS candidate proteins and integrating technologies in characterizing molecular dynamics (Case et al, 2019; Ma et al, 2021, 2022; Sun et al, 2021), protein structure, and protein–protein interactions (Ma et al, 2022; Spegg et al, 2023; Tran et al, 2020) makes rational forecasting of phase separation via computational approaches feasible.

Here, we introduce MolPhase, a protein PS and molecular condensation predictor. MolPhase was trained using 606 experimental-derived PS proteins. MolPhase applies a broad set of physicochemical features and incorporates larger and more diverse experimental datasets, which showed improved accuracy and reliability among several available phase separation-prediction algorithms, including DeePhase (Saar et al, 2021), PSPredictor (Chu et al, 2022), FuzDrop (Hatos et al, 2022), PSPer (Orlando et al, 2019). MolPhase enables efficient analysis of extensive protein sequence datasets, facilitating the identification of novel phase-separating proteins and their functional roles. By combining with the features analysis generated by MolPhase, rationale design could be carried out to dissect the underlying PS contributing factors. Using MolPhase, we found that several physical-chemical interaction modes, including pi–pi interaction, disorder ranked as top contributing features.

To assess and validate MolPhase's performance, we investigated bacterial type III effectors (T3Es) both in vitro biochemically and

[1]School of Physical and Mathematical Sciences, Nanyang Technological University, 637371 Singapore, Singapore. [2]School of Biological Sciences, Nanyang Technological University, 637551 Singapore, Singapore. [3]Institute for Digital Molecular Analytics and Science, Nanyang Technological University, 636921 Singapore, Singapore. [4]These authors contributed equally: Qiyu Liang, Nana Peng. [✉]E-mail: yansongm@ntu.edu.sg

in vivo within living host cells. T3Es naturally injected into plants during microbial infection via the type III secretion system (T3SS), accumulate in the host and gradually subvert host biology based on their localization, abundance, and recognition partners in a spatiotemporal manner. Given the known phase-separating nature of T3E XopR and the prevalent highly structure-disordered characteristics of T3Es, we hypothesize that many other phytobacterial T3Es could form molecular condensates on the plant's plasma membrane and cytoplasm (Marin and Ott, 2014; Sun et al, 2021; Wang et al, 2022b). We analyzed all the T3Es from two most studied model phytobacteria *Xanthomonas campestris* pv. *campestris* (*Xcc*) 8004 and *Pseudomonas syringae* pv. *tomato* (*Pst*) DC3000. According to the prediction of MolPhase, we chose a few representative T3Es and experimentally examined their PS in living plants via cell imaging and biochemical behavior using recombinant proteins. Our findings revealed that these T3Es undergo PS both in vitro and in vivo, consistent with MolPhase analysis. However, the homotypic PS from recombinant proteins showed different material properties and dynamics than the condensates formed in living cells. This implies that diverse microenvironments, characterized by distinct combinations of biophysical conditions, may impact the significance of intrinsic features in a particular environmental setting. Consequently, this tuning effect influences the interactions and assembly patterns of each protein undergoing phase separation. Furthermore, we conducted a global analysis of the *Xcc* 8004 and *Pst* DC3000 proteomes and compared them with the *Pseudomonas syringae* Type III Effector Compendium (PsyTEC), which contains 529 T3Es (Laflamme et al, 2020). Compared to the entire phytobacteria proteome, T3Es *Pseudomonas* species exhibit a much higher PS propensity, indicating the potential universal scaffolding mechanism by which bacterial T3Es subvert host biology through their abilities in multivalent interactions and phase separation.

# Results

## Feature characterization for machine-learning predictor

To expand the pool of protein sequences that undergo phase separation (PS) as training data, we retrieved sequences from public databases: LLPSDB (Wang et al, 2022a), PhaSePro (Mészáros et al, 2020), DrLLPS (Ning et al, 2020), PhaSepDB (Hou et al, 2023) and CD-CODE (Rostam et al, 2023), along with additional 99 sequences curated from literature manually. For all the above-mentioned sequences, only those with experimental evidence that can undergo homotypic PS were selected to construct the final training set (Dataset EV1). To remove the redundant sequences with high similarity, CD-HIT was applied to filter sequences with an identity higher than 0.9 (Fu et al, 2012). In total, we obtained 606 sequences for positive PS dataset (designated as POS) to study PS protein property and to model training in the subsequent step. Proteins deposited in the Protein Data Bank (PDB) are typically derived from monodispersed and homogenously packed proteins characterized by stable intramolecular interactions and weak intermolecular associations. While we cannot entirely dismiss the possibility of well-folded proteins undergoing phase separation through surface multivalent weak interactions under specific experimental conditions, as observed in the case of BSA in 20% PEG-8000 (Bullier-Marchandin et al, 2023) and lysozyme at concentrations exceeding 300 µM (Pyne and Mitra, 2022). Nevertheless, PDB is the current best available choice for PS-negative dataset (see "Discussion"). Here we selected 1362 proteins from PDB (Berman et al, 2000) as the negative PS dataset (designated as NEG), all training set sequences were included in Dataset EV1.

To characterize the features that contribute to PS, here we dissect multiple biophysical and biochemical properties between POS and NEG (Fig. 1A–L; Table EV1). Compared with NEG, POS contains longer sequences (Fig. 1A). The increasing in sequence length can reduce the entropic cost of confining the protein in a dense phase (Martin and Mittag, 2018). Given that PS is driven by the multivalent interaction of different domains, we explored the characteristics and discrimination of intrinsically disordered regions (IDRs) and low complexity regions (LCRs) in protein sequences from POS and NEG (Fig. 1B–D). POS contains a higher percentage of IDR and LCR (Fig. 1B,C). This is consistent with the enriched interacting motifs within IDR and LCR that serve as the "stickers", which drive molecular condensation in combination with "spacer" (Martin et al, 2020), including prion-like domain (PLD), type II polyproline helices (PPII) (Brown and Zondlo, 2012), pi interaction (including pi–pi/pi–cation interaction) (Vernon et al, 2018), charge-charge interaction, and the hydrophobic effect. Figure 1E–G shows that POS contains a higher fraction in interactive domains. In contrast, we found that fraction of charged residues (FCR) is lower in POS (Fig. 1H), and net charge per residue (NCPR) is closer to zero in POS (Fig. 1I). In addition, we introduced kappa and omega to indicate the distribution of charged residues pattern in the whole sequence. In detail, kappa value reflects the mixture of charged amino acid within the whole sequence (Das and Pappu, 2013), and omega reflects the mixture of charged residues plus proline in the sequence (Martin et al, 2016). Lower kappa and omega values indicate a superior mixture, whereas higher values suggest the formation of more localized blocks. It is noteworthy that charge patterns exhibited a better mix in NEG (Fig. 1J). However, when proline and charge amino acids were combined, there was no significant difference (Fig. 1K). When considering FCR, NCPR, and kappa together (Fig. 1H–J), it appears that proteins prone to PS generally maintain a neutral charge. Yet, they present more localized charge blocks, as indicated by kappa. The reason why omega does not show a significant difference between POS and NEG (Fig. 1K) remains unclear. This suggests that most proteins adapt to the physiological cellular neutral pH condition without inherent homotypic PS. Instead, they facilitate inter- and intramolecular interactions using localized charge blocks and other interactive motifs as needed. Regarding hydrophobicity, POS is more hydrophilic than NEG (Fig. 1L). Interestingly, while local hydrophobic cluster/domain could enhance coacervation (Yeo et al, 2011) and stabilize phase-separating FUS (Krainer et al, 2021), a lower overall ratio of hydrophobic residues is believed to maintain amino acid chains in a disordered states, potentially facilitating condensation in more liquid-like state (Dignon et al, 2020). This implies that the role of hydrophobicity in guiding molecular condensation requires protein-specific and position-based consideration. In our findings, POS has a lower hydrophobicity than NEG, whereas charge blocks directly drive PS (Fig. 1E–L) (Lyons et al, 2023; Pak et al, 2016).

We categorized amino acids into distinct clusters based on their biochemistry properties (Table EV2). Our findings revealed that

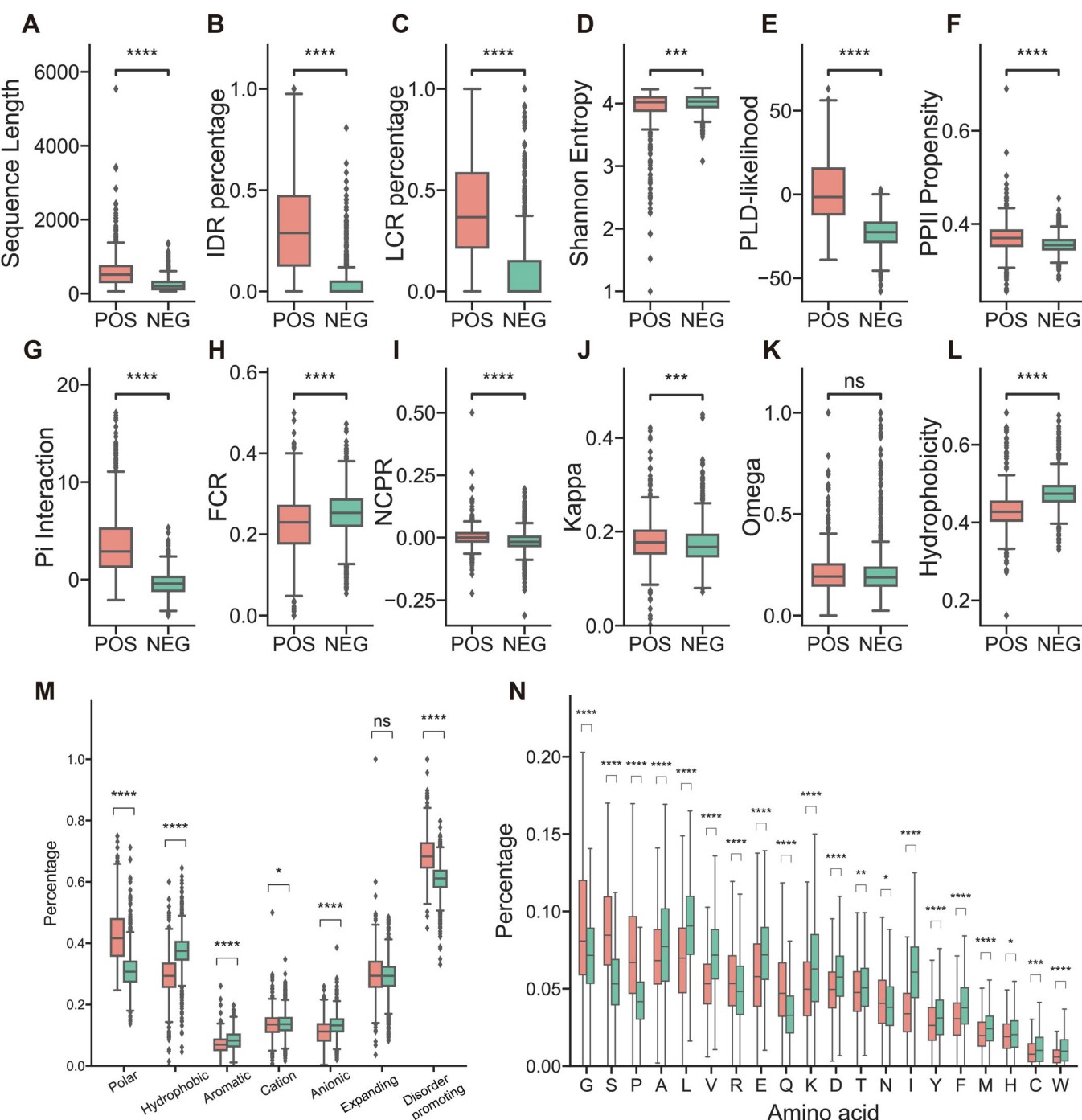

**Figure 1. Features comparison of phase separation positive (POS) and negative (NEG) proteins in the training dataset.**

The comparison includes (**A**) Sequence length, (**B**) IDR percentage, (**C**) LCR percentage, (**D**) Shannon Entropy, (**E**) Prion-like domain log likelihood, (**F**) Type II Polyproline helices propensity, (**G**) pi interaction, (**H**) fraction of charged residues, (**I**) net charge per residue, (**J**) kappa, (**K**) omega, and (**L**) hydrophobicity. In addition, (**M**) demonstrates a percentage comparison of different groups of amino acids, and (**N**) showcases a single amino acid percentage within full-length protein sequence. Features from (**H–M**) are defined by localCIDER. Data information: (**A–N**) The central line illustrates the median value, while the edges of the box denote the 25th (lower) and 75th (upper) percentiles. The whiskers extend to 1.5× the interquartile range from the box edges. Outliers beyond the whiskers' range are marked as dots. For $n = 606$ in POS and $n = 1362$ in NEG. Statistical significance is indicated as *$P \leq 0.05$, **$P \leq 0.01$, ***$P \leq 0.001$, ****$P \leq 0.0001$, and ns not significant (Mann–Whitney test, two-tailed for (**A–N**)). Source data are available online for this figure.

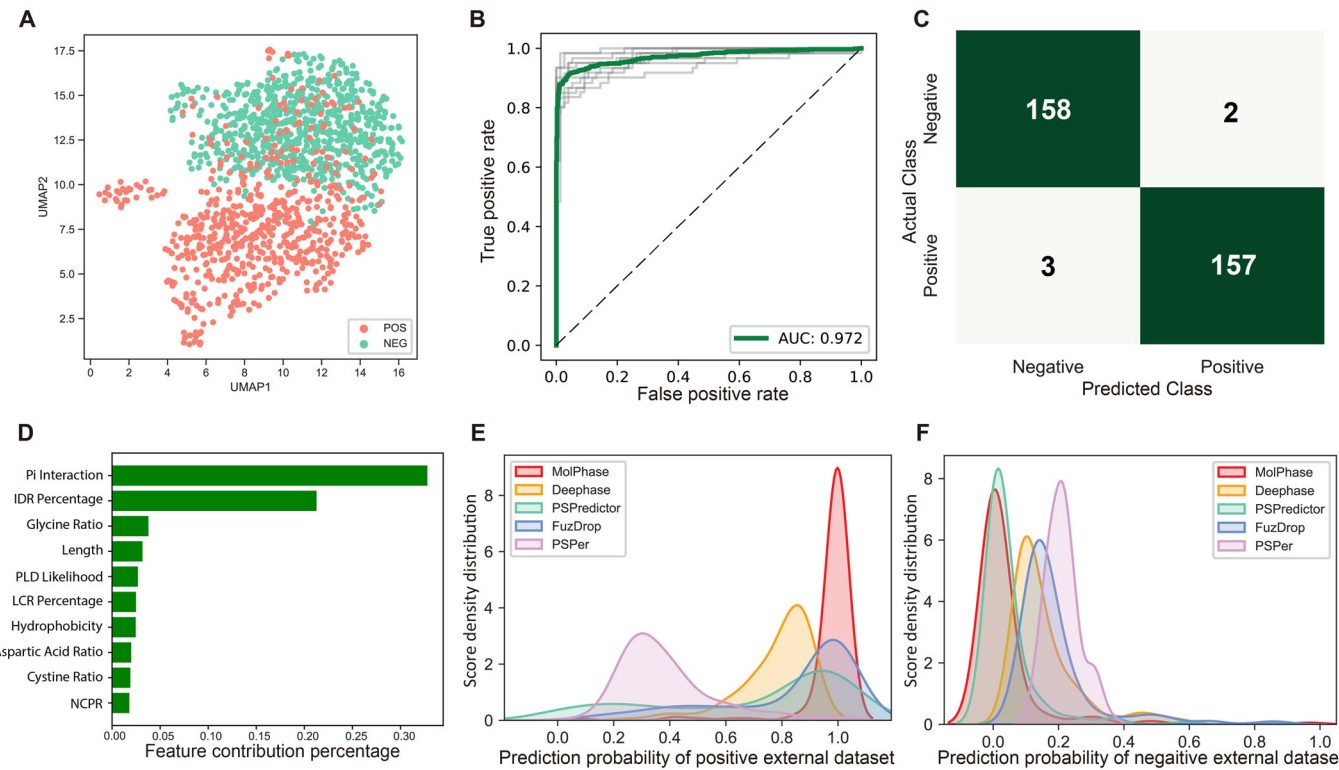

**Figure 2. Performance evaluation of the machine-learning predictor, molecular condensation, and phase separation predictor (MolPhase).**

(A) UMAP visualization of the 2D vector projection of training datasets following one-sided selection undersampling. (B) Receiver operating characteristic (ROC) curve for the classifier using tenfold cross-validation. The green line depicts the average of all training outcomes, while the gray lines illustrate the ROC curves from ten training iterations. The area under the curve (AUC) is 0.972. (C) Confusion matrix showcasing predictions for the external dataset made by MolPhase. (D) A representation of the ten most pivotal features in MolPhase. The x axis denotes feature importance. (E) Density plot of the prediction probability for the external positive dataset, and (F) negative dataset, as generated by five distinct predictors. On the x axis is the prediction probability of proteins with phase separation propensity as determined by each predictor, while the y axis displays the density. Source data are available online for this figure.

PS-prone proteins are rich in polar and disorder-promoting amino acids, yet they have fewer aromatic and anionic residues (Fig. 1M). As an illustration, amino acids like glutamine and asparagine, which are enriched in PLD (Paul and Ross, 2015) and those that facilitate the formation of disordered regions, such as glycine and serine (Campen et al, 2008) exhibit higher levels in POS. Conversely, hydrophobic amino acids like alanine and valine are more prevalent in NEG (Fig. 1M). Furthermore, by considering each amino acid as an individual feature and examining the percentage distribution of amino acids between the POS and NEG datasets (Fig. 1N), we discovered a composition bias between POS and NEG. This additional insight enhances the power of combinational feature analysis and classification through amino acid information (Fig. 1N,M).

## Machine-learning model construction and feature evaluation

We selected 39 features, which were all the known intrinsic phase separation-related physicochemical features (Fig. 1). By employing the uniform manifold approximation and projection (UMAP) algorithm (McInnes et al, 2018), we transformed this 39-dimensional data into a 2-dimensional scatter plot. Even without applying a clustering algorithm, a clear distinction between POS

and NEG sets is evident, with both forming two separate groups (Fig. EV1B). This suggests that the chosen features effectively capture the difference between POS and NEG. To improve the performance of final model, we used the one-sided selection algorithm (Lemaître et al, 2017) to undersample the NEG set, ensuring a balanced training dataset. After undersampling, we again used UMAP to visualize the 2D projection, and two sets remain to be distinctly separated (Fig. 2A).

Following the workflow depicted in Fig. EV1A, we tested seven different classifier models on the same training set (Fig. EV1C). Among these, extreme gradient boosting (XGBoost) (Chen and Guestrin, 2016) stood out, delivering the best performance (Fig. EV1C). Consequently, we selected XGBoost as the final model. We assessed the model's efficacy using a 10-fold cross-validation and plotted the receiver operating characteristic (ROC) curve. The area under the curve (AUC) was an impressive 0.972 (Fig. 2B). For further validation, we employed external datasets to evaluate the performance of our predictor. We utilized the negative testing datasets from DeePhase (Saar et al, 2021) and sourced positive and negative PS sequences from PhaSepDB (You et al, 2020) and PDB, respectively. To enhance the testing reliability, we excluded sequences from the testing set that bore high similarity to those in the training set. Specifically, we removed the positive testing set with an identity greater than 0.9 and from the negative

testing set with an identity above 0.3. The resulting confusion matrix revealed that our model achieved a true positive rate of 98.1% and a true negative rate of 98.8% on the external testing set (Fig. 2C). Only five proteins were wrongly classified within 320 proteins in total. Further analysis of feature contribution uncovered that pi interaction has the most significant effect on molecular condensation (Fig. 2D). This is followed by IDR percentage. Interestingly, glycine's ratio accounted for 3.78% in our final model, making it the third most important feature. Previous studies have indicated that glycine residues can enhance fluidity in the PS protein (Wang et al, 2018). Here, we named our predictor the Molecular Condensation and Phase Separation Predictor (MolPhase).

To evaluate the efficacy of MolPhase, we used the same testing sets to juxtapose its performance with four other previously published predictors: DeePhase (Saar et al, 2021), PSPredictor (Chu et al, 2022), FuzDrop (Hatos et al, 2022) and PSPer (Orlando et al, 2019). When compared with these predictors using our positive phase separation protein set, MolPhase displayed the lowest false negative rate (Figs. 2C and EV1D–G). Its predicted scores were predominantly clustered around 1 and did not exhibit the pronounced "long tail" effect seen in others (Fig. 2E). On the negative test set, all five predictors showed similar accuracy rates (Figs. 2C and EV1D–G). However, scores from MolPhase and PSPredictor (Chu et al, 2022) were predominantly closer to 0 (Fig. 2F). In summary, our data underscores MolPhase's ability to accurately predict both PS positive and negative proteins to provide appropriate weighing in homotypic condensation.

## Application of MolPhase for evaluating phytobacterial effectors' phase separation in plant host

We sought to validate the MolPhase predictions using a specific biological system. or this purpose, we examined the type III effector (T3E) of pathogenic bacteria both in vitro and within host cells. Phytopathogens bacteria utilize the type III secretion system (T3SS) to progressively inject a diverse array of T3E into plant cells throughout the infection process. These T3Es, in a dose-dependent manner, suppress the host's immune defense, facilitating pathogen's growth (Jones and Dangl, 2006). Due to the need to navigate through the narrow, needle-like T3SS, T3Es often possess intrinsic properties of high disorder, enhancing their conformational flexibility (LeBlanc et al, 2021), which, on the other side, also make them ideal candidates for molecular condensation. We subsequently scrutinized all the T3Es from two extensively researched phytopathogenic model species, *Pseudomonas syringae* pv. *tomato* (*Pst*) DC3000 and *Xanthomonas campestris* pv. *campestris* (*Xcc*) 8004, based on their annotated genomes (Buell et al, 2003; Qian et al, 2005). Upon assessing the molecular condensation likelihood of these T3Es, our previously identified phase separation T3E, XopR, emerged as the top contender (Sun et al, 2021) (Fig. 3A). For further validation, we randomly selected eight T3Es - HopS1, HopA1, HopAB2, HopO1-2, XopD2, XopX2, XopAH, and XopAY (Figs. 3B,C and EV2A–F)— representing a range of MolPhase propensities. Analyses of their biophysical and biochemical features through MolPhase, combined with structure predictions from AlphaFold2, were shown together (Figs. 3B–E and EV2A–D). Notably, while the IDR is important for PS

(Gao et al, 2022), there's a stark contrast between the IDR percentages of chosen effectors, HopS1 and HopA1, at 80.5% and 8.9% (Fig. 3B,C), respectively. Moreover, we also incorporated five proteins predicted to be negative, XopQ, mRuby2, HopE1, HopQ1-2, and XopAZ for further examination (Figs. 3D,E and EV2G–I).

Next, we proceeded to express and isolate the recombinant proteins of three T3Es and mRuby2 using the *Escherichia coli* system. Securing full-length proteins that are high in intrinsically disordered regions (IDRs) and prone to condensation can be challenging in a few cases due to factors like degradation or precipitation (Alberti et al, 2018). However, the full-length HopS1 protein was successfully obtained by co-expressing with its native chaperone, ShcS1, which is located in the same operon as HopS1 (Kabisch et al, 2005). The other effector proteins employed in this study were purified using a His-SUMO construct (see "Methods"). In vitro, HopS1 and HopA1 exhibited typical phase separation behavior, dependent on protein concentration and ionic strength (Fig. 4A,B). Conversely, the MolPhase-predicted negative XopQ failed to form distinct spherical droplets under the conditions tested, forming only sparse tiny punctuates that did not significantly change in size over time (Fig. 4C). Another negative candidate, mRuby2, maintained a consistent diffused distribution in solution across all conditions tested (Fig. 4D). Notably, while both HopS1 and HopA1 formed spherical droplets, HopS1 displayed higher fluidity than HopA1 in an in vitro fluorescence recovery after photobleaching (FRAP) assay. We then assessed their in vivo phase behavior by expressing them in *Nicotiana benthamiana* via an agrobacterium-based transient expression system, controlled by an inducible XVE promoter responsive to β-estradiol (Zuo et al, 2000). Both HopS1 and HopA1 exhibited two-dimensional condensation on the plasma membrane (PM), in contrast to XopQ and mRuby2, which dispersed throughout the cytosol (Fig. 4G). A few punctuated signals persist in XopQ across various concentrations, indicating the presence of a minor non-phase-separating oligomer packing species (Miao et al, 2023). Intriguingly, within plant cells, both HopS1 and HopA1 displayed greater dynamics than their in vitro homotypic condensates, as evidenced by FRAP (Fig. 4E,F,H,I). This suggests a balanced interplay of inter- and intramolecular interactions in vivo, possibly preventing the formation of higher-order and denser assemblies seen in vitro. Indeed, a wide range of physicochemical conditions contributes to the multivalent interactions that underlie phase separation. These conditions include the presence of small molecules (Zavaliev et al, 2020), ROS species (Huang et al, 2021), pH variations (Adame-Arana et al, 2020), changes in ionic strength (Sun et al, 2021), and conditions of molecular crowding (Delarue et al, 2018). We additionally evaluated six other effectors predicted to be positive in vivo in plants. HopO1-2 exhibited 2D phase separation on the PM, whereas HopAB2, XopD2, XopX2, XopAH and XopAY formed condensates within the cytosol (Fig. 4G). Besides that, three effectors HopA1, XopAH and XopAY which can form in vivo condensation, were predicted to be phase separation negative by most of other predictors, which further demonstrated the outstanding performance of MolPhase (Fig. 3A). In conclusion, all the T3Es examined that were positively predicted by MolPhase displayed phase separation behaviors within live plants upon exogenous expression, mimicking their deposition into the host by T3SS during plant–microbe interactions (Fig. 3A).

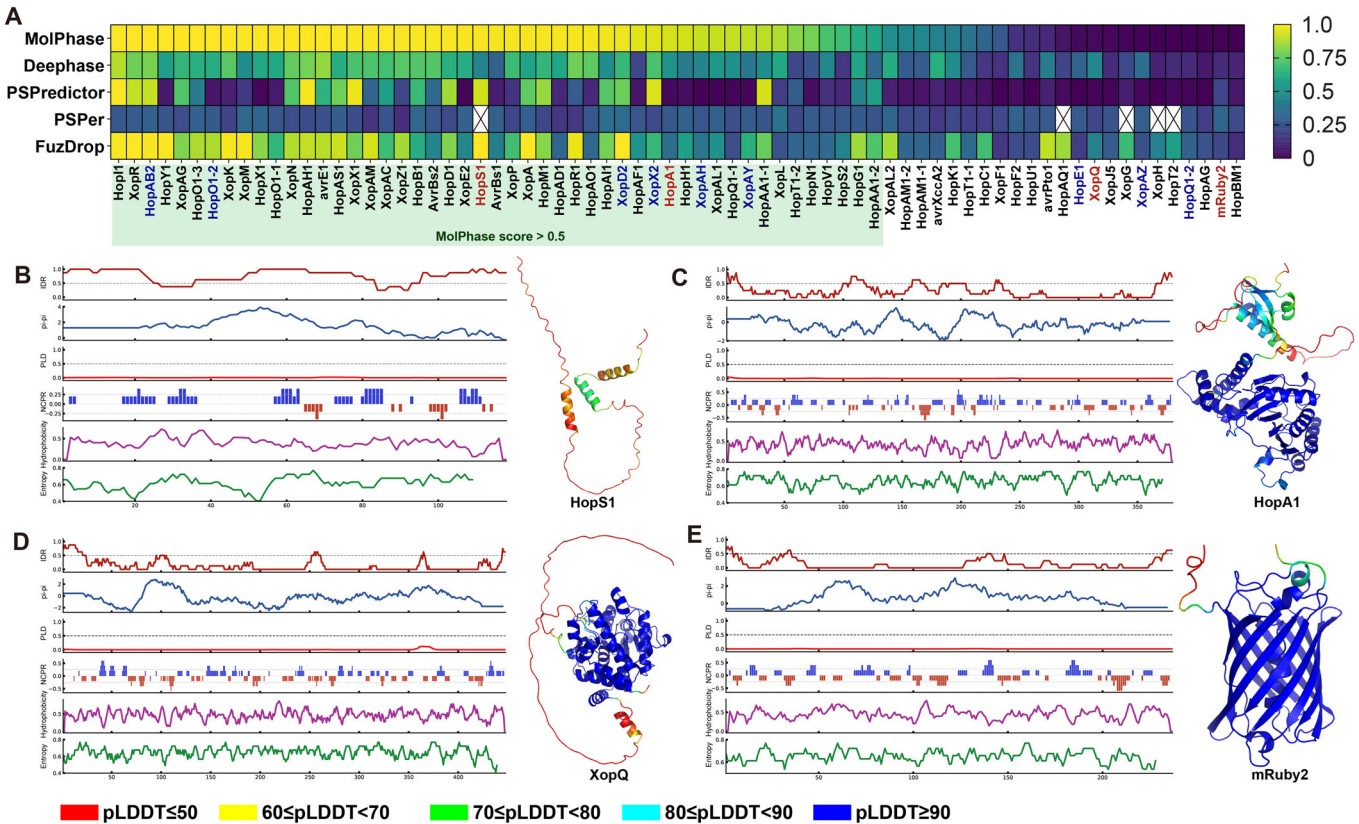

**Figure 3. Extraction of features and structural prediction of effector proteins.**

(A) Prediction of phase separation propensity for effector proteins from *Xanthomonas campestris* pv. *campestris* (*Xcc*) 8004, *Pseudomonas syringae* pv. *tomato* (*Pst*) DC3000, and the fluorescent protein mRuby2. Five effectors could not be predicted by PSPer for unspecified reasons and are absent in the heatmap. Effectors are arranged based on the MolPhase score in descending order. Extraction of features and AlphaFold2 structural prediction for two positive candidates: (B) HopS1 and (C) HopA1, and two negative candidates: (D) XopQ and (E) mRuby2. The feature plot sequence from top to bottom is IDR, pi interaction, prion-like domain likelihood, net charge per residues, hydrophobicity, and Shannon entropy. AlphaFold2-predicted structures are color-coded by the predicted local-distance difference test (pLDDT) show at the bottom of image, with structures having a higher pLDDT indicating higher accuracy. Regions with low pLDDT, especially lower than 50 strongly tend to disorder. The size of the structure images is not to scale. Source data are available online for this figure.

## Phase separation under proteomic perspective

Taking advantage of MolPhase's high-throughput capability to predict phase separation (PS) at the proteomics level in living organisms, we analyzed the entire proteomes of *Xcc* 8004 and *Pst* DC3000. Remarkably, the distribution patterns of protein PS in these two strains were quite similar. The majority of proteins exhibited no potential for PS, while only a few proteins demonstrated the ability to undergo PS (Fig. 5A,B). We then compared the PS distribution of the phytobacterial proteome with that of their T3Es. To provide a comprehensive evaluation of phytobacterial T3Es, we expanded our analysis to include 529 effectors from 494 strains sourced from *Pseudomonas syringae* Type III Effector Compendium (PsyTEC) (Laflamme et al, 2020). As a result, this extensive set of T3Es showed a higher propensity to undergo PS (Fig. 5C), with 63.5% of them predicted to have a MolPhase score exceeding 0.5. The top six features that most influenced the PsyTEC set prediction (Figs. 2D and 5D–I) are consistent with MolPhase's training datasets (Fig. 1B,C,E,G,L,M), suggesting that T3Es have a similar amino acid composition and sequence features to the PS proteins from various species.

To derive functional insights from MolPhase-predicted proteins, we identified proteins with prediction scores above 0.9 and below 0.1 in the proteomes of *Pst* and *Xcc*. These were designated as phase positive and negative sets, respectively, and then subjected to pathway enrichment analysis (Figs. 5J,K and EV3A,B). Kyoto Encyclopedia of Genes and Genomes (KEGG) enrichment for positive PS proteins revealed several pathways associated with nucleic acids, including RNA degradation, mRNA biogenesis, DNA repair, and recombination (Fig. 5J,K). Intriguingly, these nucleic acid-binding protein candidates are recognized hotspots for PS (Harami et al, 2020; Kar et al, 2022), aligning with their tendency to form complex condensates with long-chain polyelectrolyte DNA/RNA. This also underscores the potential of DNA/RNA as a versatile interaction platform for their associated proteins. Thus, such interactions, including IDR-IDR associations, could enhance the formation of protein-nucleotide complex PS (Shin and Brangwynne, 2017). Notably, many T3Es from T3SS, categorized under "secretion system", also showed a propensity for PS in the proteomes of *Xcc* and *Pst*. On the other hand, MolPhase's predicted PS-negative proteins highlighted several consistent pathways in KEGG enrichment, like amino acid metabolisms and ABC transporter. Interestingly, two nucleic acid-related pathways, particularly tRNA

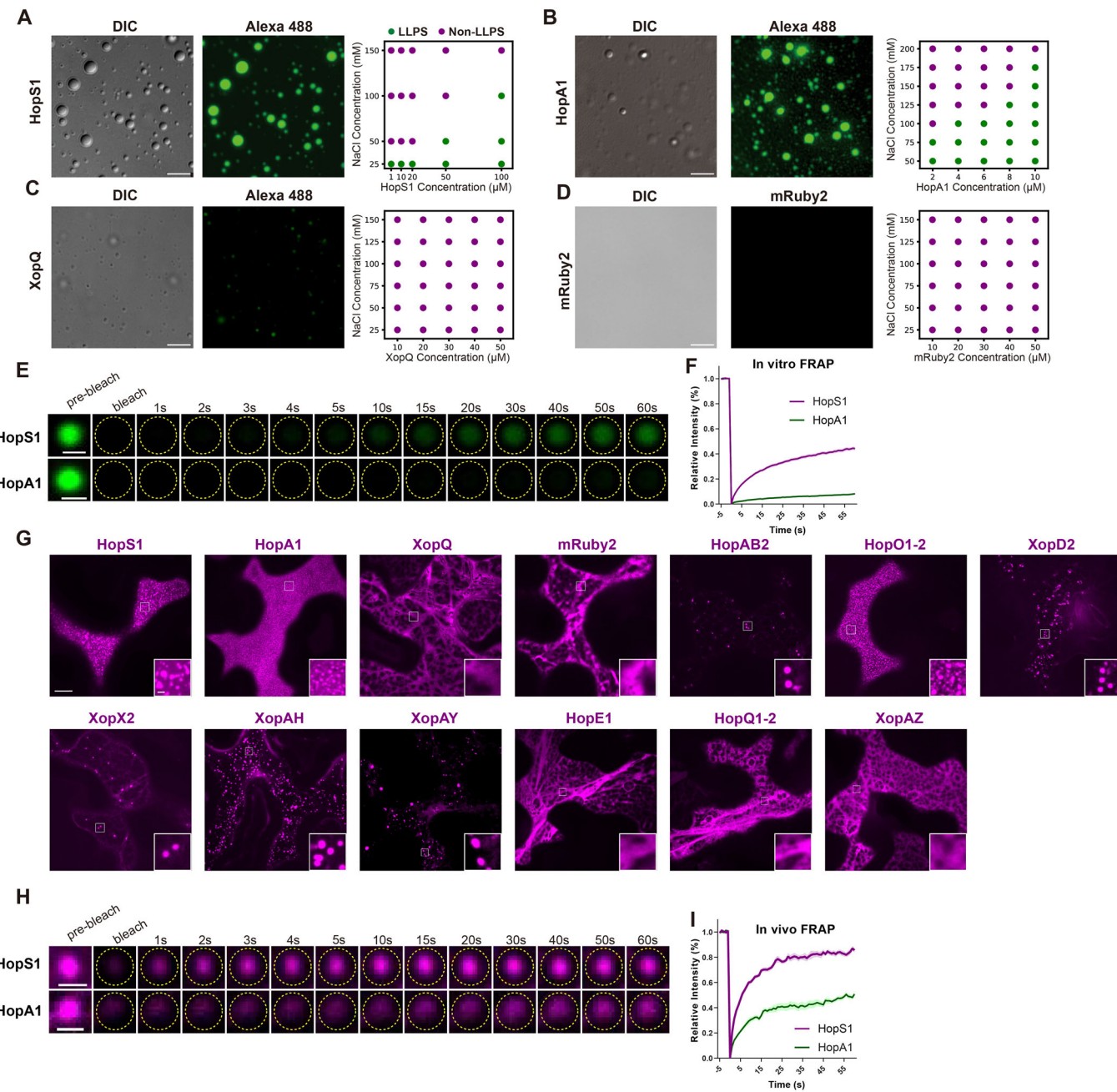

**Figure 4. Phase behavior of examined proteins both in vitro and in vivo.**

(A–D) In vitro phase separation studies of proteins predicted for phase separation: (A) HopA1, (B) HopS1, and those predicted against phase separation: (C) XopQ, (D) mRuby2. 1% Alexa 488-labeled protein was combined with 99% unlabeled protein for HopS1, HopA1, and XopQ. Microscope conditions were: (A) 100 μM HopS1 with 25 mM NaCl, (B) 50 μM HopA1 with 50 mM NaCl, (C) 50 μM XopQ with 25 mM NaCl, and (D) 50 μM mRuby2 with 25 mM NaCl. Images have a scale bar of 5 μm. The phase diagram indicates conditions favorable or unfavorable for liquid–liquid phase separation (LLPS). (E) In vitro FRAP assay for HopS1 and HopA1 with a scale bar of 1 μm. (F) Relative intensity analysis of the in vitro FRAP assay for HopS1 and HopA1. Intensity prior to bleaching (5 s) was averaged and set to 100%, post-bleach intensity was set to 0%, and intensity 1 min post-bleach was recorded and normalized. (G) In vivo tobacco transient expression studies of eight proteins predicted for phase separation: HopS1, HopA1, HopAB2, HopO1-2, XopD2, XopX2, XopAH, XopAY, and five predicted against it: XopQ, mRuby2, HopE1, HopQ1-2 and XopAZ. HopS1, HopA1, HopAB2, HopO1-2, XopD2, XopX2, XopAH, XopQ, mRuby2, HopE1, and HopQ1-2 were driven by inducible XVE promoter, while XopAY and XopAZ were driven by 35 S promoter. Scale bars are 10 μm for broader images and 1 μm for close-ups. (H) In vivo FRAP assay for HopS1 and HopA1. Scale bar = 1 μm. Yellow deshed cirules in (E, H) indicated the FRAP area. (I) Relative intensity analysis of the in vivo FRAP assay for HopS1 and HopA1, using the same intensity calculation methods as in (F). Data information: Data in (F, I) represent ten independent biological repeats. The central bold line indicates the mean value, while the shaded areas represent standard deviation. Source data are available online for this figure.

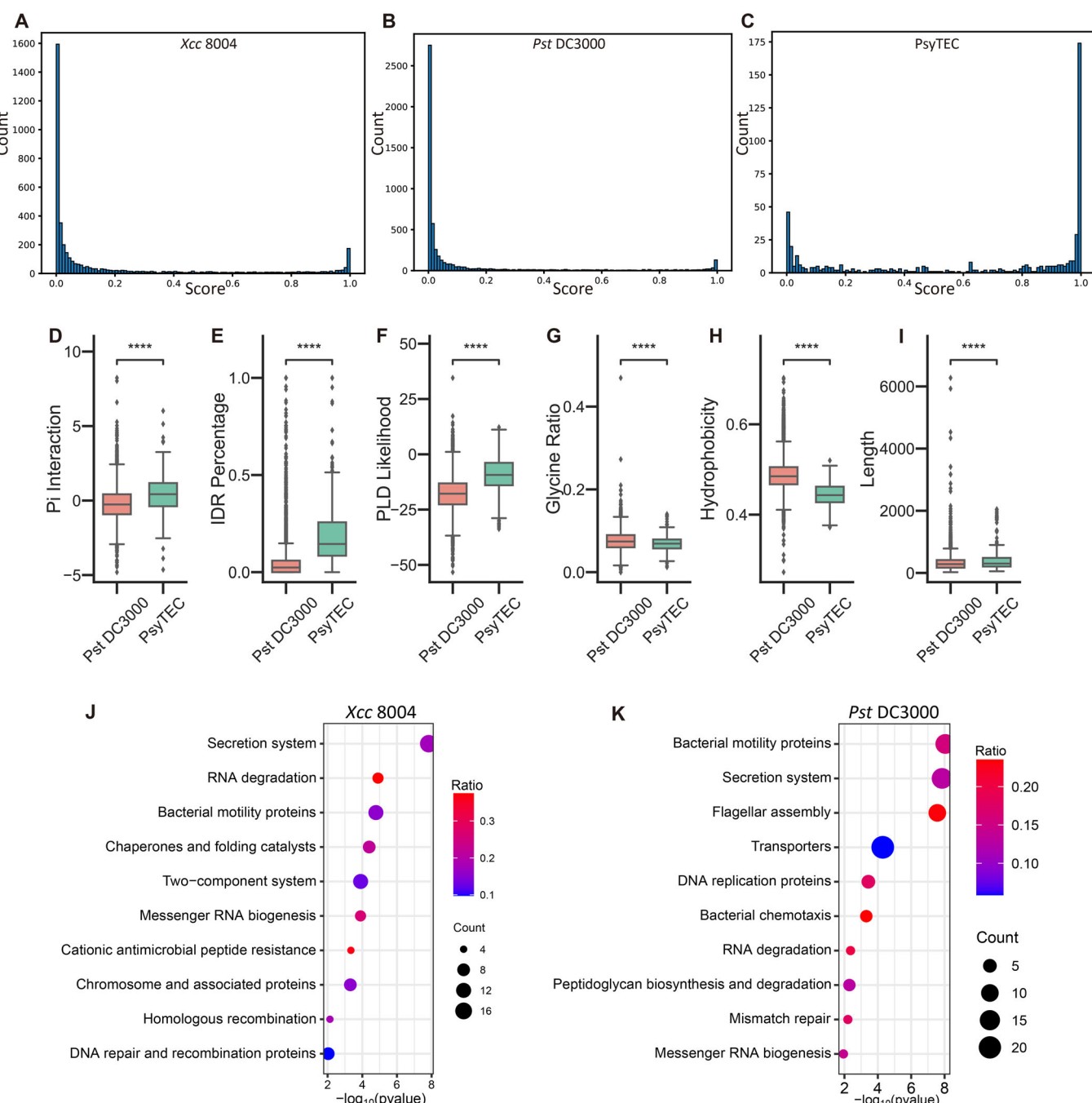

**Figure 5. Distribution of phase separation proteins in the proteome.**

(**A**, **B**) Predicted phase separation across the entire proteome for (**A**) *Xcc* 8004 and (**B**) *Pst* DC3000. (**C**) Phase separation prediction for 529 *Pseudomonas* effectors from the *Pseudomonas syringae* Type III Effector Compendium (PsyTEC), with the x axis showing prediction scores in 0.01 increments and the y axis indicating quantity. (**D–I**) Feature comparisons between the *Pst* DC3000 proteome and PsyTEC, including (**D**) Pi interaction, (**E**) IDR percentage, (**F**) PLD-likelihood, (**G**) Glycine ratio, (**H**) hydrophobicity, and (**I**) sequence length. (**J**, **K**) Kyoto Encyclopedia of Genes and Genomes (KEGG) enrichment analysis for proteins with phase separation prediction scores above 0.9 in (**J**) *Xcc* 8004 and (**K**) *Pst* DC3000, where the x axis shows the $-\log_{10}$ P value for the Fisher exact test, the y axis lists enrichment items, the ratio indicates the percentage of the entire pathway enriched, and the count shows the number of enriched items. Data information: In boxplots from (**D–I**), the central line denotes the median, box edges mark the 25th and 75th percentiles, whiskers extend to 1.5× the interquartile range from the box edges, and outliers are dots. For $n = 5417$ proteins in *Pst* DC3000 and $n = 529$ effectors in PsyTEC. Significance levels are indicated as ****$P ≤ 0.0001$ (Mann–Whitney test, two-tailed in (**D–I**)). Source data are available online for this figure.

biogenesis and ribosome, primarily consisted of predicted PS-negative proteins in both *Xcc* and *Pst* proteomes. The precise role of nucleic acids in PS requires more in-depth analysis.

## Online interface for MolPhase

To make MolPhase publicly available and easily accessible, we have developed an online interface (http://molphase.sbs.ntu.edu.sg/) (Fig. EV4). This platform is constructed using the 39 integrated features (Fig. 1). When users input a protein sequence of interest, MolPhase returns a probability score for phase separation (PS) that ranges from 0 to 1. We recommend 0.5 as an initial cutoff to evaluate the queried protein. For the online version of MolPhase, it allows the protein sequence input range from 40 to 3000 amino acids without invalid characters. Furthermore, MolPhase offers a distribution of features along the sequence, assisting users in functional evaluation and hypothesis formulation. This is achieved through a comprehensive sequence scan and a parallel comparison of protein folding and associative motifs, catering to both homotypic and heterotypic interactions. With the insights provided by MolPhase's detailed analysis and PS predictions, the design of targeted experiments and subsequent functional explorations will be significantly expedited.

# Discussion

## Constructing a higher accuracy phase separation predictor

Biomolecular condensation is primarily driven by a myriad of interaction forces, including hydrophobicity, electrostatic pi/cation-pi, dipole-dipole, and protein–protein/nucleic acid interactions, etc. (Boeynaems et al, 2018). Various motifs and regions, equipped with inter- and intramolecular interactions, facilitate the initial association that leads to the formation of higher-order assemblies, such as IDR, PLD, oligomerization motifs, nucleic acid-binding motifs, etc. Initial generation of phase separation predictors primarily focused on individual interaction forces and incorporated only single or a few features (Vernon and Forman-Kay, 2019), such as PLAAC (Lancaster et al, 2014) was based on PLD prediction and PScore (Vernon et al, 2018) based on pi–pi contacts. Often an engagement of fewer features leads to suboptimal performance, especially when compared to predictors that integrate multiple features (Chen et al, 2022). To offer a more holistic perspective on phase separation, we present MolPhase. It integrates a total of 39 features, making it the most comprehensive tool that addresses the array of interaction forces driving phase separation to date. The superior performance of MolPhase, as depicted in Fig. 2E,F, is likely attributed to its ability to factor in more relevant phase separation interactions than other predictors (Chu et al, 2022; Hatos et al, 2022; Orlando et al, 2019; Saar et al, 2021). The correlation coefficient within different features also indicated the importance of incorporating multiple features, as the low correlation suggests that a single feature cannot adequately represent the overall characteristics of the datasets (Fig. EV1H–I). While pi-interactions remain a pivotal factor in MolPhase (Fig. 2D), solely relying on pi–pi contacts (Vernon et al, 2018) for predicting phase separation often results in reduced accuracy (Chen et al, 2022), further emphasizing that phase separation is influenced by various interactions. Narrowing down to only a subset of them might compromise prediction accuracy.

There are now several high-quality databases specifically for phase separation proteins (Hou et al, 2023; Li et al, 2020; Mészáros et al, 2020; Ning et al, 2020; Rostam et al, 2023). We have harnessed the most recent information on phase separation to build our predictor. By curating the experimentally validated homotypic PS protein, we ensured the quality for input data. And we also achieved the largest training set in terms of scaffold PS protein as our known. As these databases continue to grow, we anticipate that the accuracy of phase separation predictors will enhance correspondingly. However, concerning negative training and testing datasets, such as DeePhase (Saar et al, 2021), PSPredictor (Chu et al, 2022), FuzDrop (Hardenberg et al, 2020), including ours, still rely on sequences from PDB. As PS is conditional-based, some globular protein deposited in PDB could still undergo PS within certain conditions. As mentioned above, BSA in 20% PEG-8000 (Bullier-Marchandin et al, 2023) and lysozyme at concentration higher than 300 μM (Pyne and Mitra, 2022) could undergo phase separation. In general, these globular folded proteins need extreme conditions to undergo PS, and their in vivo biological functions are non-PS relevant (Boeynaems et al, 2018). If all these extreme conditions needed PS protein were inputted into the positive training set, tremendous false positive data will be output as prediction results.

On the other hand, although disorder is long to thought to be one of the most important characteristics of PS, disorder itself could not drive PS directly (Boeynaems et al, 2018). Under the "sticker-spacer" framework, interaction forces such as pi interaction, charge blockage inside IDR could serve as contribution factors, and the disorder could provide flexibility (Martin et al, 2020). To better evaluate these interaction forces, we integrated them into the MolPhase (Fig. 1), and experimental validation results showed that MolPhase could capture PS protein even with low IDR percentage (Fig. 4), indicating the limitation in using IDR for descripting phase-separating proteins. Currently, IDR-centric model was reported that analyzed biophysical features inside budding yeast IDR (Zarin et al, 2021). Here, from analyzing 606 experimentally validated phase-separating proteins, we found that IDR showed diverse correlation with other biophysical features, either positive or negative, or spanning a broad spectrum in correlation coefficient (Fig. EV1H,I).

The PDB mainly catalogs well-structured proteins or domains, possess low conformational entropy, which distant from the PS protein have high entropy (Fuxreiter and Vendruscolo, 2021), make them as the good candidates for negative training/testing set in terms of near-native conditions. However, owing to the limitation of protein crystallization, items in PDB often omitting disordered. Lack of contain long IDR but PS-negative training data might dampen the MolPhase performance. But all in all, PDB is the best choice for phase separation negative dataset at this point. Thus, utilizing higher-quality negative datasets are likely to further improve future PS predictors.

## From MolPhase prediction to condensates properties characterization and potential biological functions

Predicting protein phase separation using sequence information hinges on understanding the biophysical attributes of the protein. This includes its charge, hydrophobicity, and amino acid composition, as they can determine the likelihood of the protein undergoing phase separation. However, the process of protein phase separation is highly dynamic and subject to their surrounding environmental factors, such as crowding (Delarue et al, 2018), concentration of binding partners (Case et al,

2019), pH (Adame-Arana et al, 2020), temperature (Boeynaems et al, 2018), ionic strength (Sun et al, 2021). These conditions can modify the interaction and structure of inherent associative motifs, making primary sequence-based predictions of material properties challenging. Therefore, researchers should exercise extra caution when directly applying MolPhase scores that are close to each other as experimental expectations, given their dependence on specific conditions. Nevertheless, juxtaposing a precise MolPhase prediction with structural predictions, such as those made by AlphaFold2 (Jumper et al, 2021), can still streamline hypothesis formulation and experimental design for both in vitro and in vivo functional studies.

Though molecular condensation highly emphasize on the rapid phase behavior switches upon reaching critical concentrations of percolation, phase separation, or density transition (Miao et al, 2023; Pappu et al, 2023), the biological function also hinges on aspects like spatial distribution, size growth, and molecule partition. These are majorly modulated by nucleation and coalescence over space and time (Lee et al, 2023). One key role of PS is to dampen cellular noise by creating concentrated, membraneless compartments (Deviri and Safran, 2021; Klosin et al, 2020; Riback and Brangwynne, 2020). To gauge a biomolecule's noise-canceling potential in various biological settings, MolPhase presents a tailored approach for predicting such behavior. Our validation of MolPhase predictions using phytobacterial T3Es aptly illustrated a plant pathology-related process. Introducing bacterial proteins into plant cells revealed that in vivo PS behavior hinges largely on the inherent attributes of the molecules being examined, without the influence of existing binding partners.

However, the widespread of low sequence complexity regions in PS protein hinder the utilization of traditional sequence alignment-based methods to characterize the biological function of disordered protein. Zarin et al (Zarin et al, 2019) found that, disordered sequences shared similar function could be clustered together by hierarchical clustering based on their sequence feature, such as kappa, omega, etc. And a new model FAIDR was developed for disordered protein function prediction (Zarin et al, 2021). Therefore, biological functions of PS protein might also be predicted by sequence feature-based clustering approaches. Dissecting the biological functions of phase-separating proteins demands a multidisciplinary approach, intertwining both computational and experimental techniques. This entails localized quantitative molecule measurements, advanced molecular assembly imaging ranging from nanometer to mesoscale, and accurate assessment of functional molecule partitioning and stoichiometry (Case et al, 2019; Dine et al, 2018; Ditlev et al, 2019; Huang et al, 2019; Hubatsch et al, 2021; Kent et al, 2020; Knerr et al, 2023; Lee et al, 2023; Su et al, 2016). To design experiments specifically for biomolecules in the context of their physiological or pathological pathways, MolPhase offers an accurate initial screen in a high-throughput manner, allowing for a systematic evaluation by integrating GO annotation, KEGG pathway enrichment analysis, or AlphaFold2-based structural prediction (Jumper et al, 2021). Beyond individual proteins, MolPhase can handle whole proteomes, pinpointing pathways abundant with phase separation-prone proteins. This underscores the significance of phase separation in those pathways. This suggests the importance of phase separation in these identified functional pathways. In our comprehensive analysis of prokaryote phytopathogens *Xcc* 8004 and *Pst* DC3000, it emerged that a majority of their proteins do not undergo phase separation and are predominantly associated with metabolism pathways. This suggests these reactions likely rely on direct binding

and precise stoichiometry to deliver quick results, rather than participating in non-linear, equilibrium-based reactions within condensates. On the flip side, the bacterial secretion pathway emerged as a prime candidate for phase-regulated pathways in both strains. In line with this, T3Es analyses of both strains, as well as the *Pseudomonas syringae* Type III Effector Compendium (PsyTEC), revealed a significant number of phase separation-prone candidates. Combining the effectors PS prediction on *Xcc* 8004, *Pst* DC3000, and PsyTEC (Figs. 3A and 5C), we shown that phytobacteria effectors are hot candidates for PS (Sun et al, 2021). And how they exploit the PS property to subvert plant immunity needs further investigation. By applying MolPhase's prowess in high-throughput predictions on a wider spectrum, the role of PS underneath evolution could be revealed.

# Methods

**Reagents and tools table**

| Reagent/resource | Reference or source | Identifier or catalog number |
|---|---|---|
| **Experimental models** | | |
| *Escherichia coli* | N/A | DH5α |
| *Escherichia coli* | N/A | BL21(DE3) Rosetta T1R |
| *Agrobacterium tumefaciens* | N/A | GV3101 |
| *Nicotiana benthamiana* | N/A | N/A |
| **Recombinant DNA** | | |
| pER10-XVE-HopS1-mRuby2 | This study | N/A |
| pER10-XVE-HopA1-mRuby2 | This study | N/A |
| pER10-XVE-XopQ-mRuby2 | This study | N/A |
| pER10-XVE-mRuby2 | This study | N/A |
| pER10-XVE-HopAB2-mRuby2 | This study | N/A |
| pER10-XVE-XopD2-mRuby2 | This study | N/A |
| pER10-XVE-HopO1-2-mRuby2 | This study | N/A |
| pER10-XVE-XopX2-mRuby2 | This study | N/A |
| pER10-XVE-XopAH-mRuby2 | This study | N/A |
| pER10-XVE-HopE1-mRuby2 | This study | N/A |
| pER10-XVE-HopQ1-2-mRuby2 | This study | N/A |
| pHGW-35S-XopAY-mRuby2 | This study | N/A |
| pHGW-35S-XopAZ-mRuby2 | This study | N/A |
| pSUMO-His-HopA1 | This study | N/A |
| pSUMO-His-XopQ | This study | N/A |
| pET-28a(+) ShcS1-HopS1-His | This study | N/A |
| pET-28a(+) mRuby2-His | This study | N/A |
| **Antibodies** | | |
| N/A | N/A | N/A |
| **Oligonucleotides and other sequence-based reagents** | | |
| PCR primers | This study | Table EV3 |

| Reagent/resource | Reference or source | Identifier or catalog number |
| --- | --- | --- |
| **Chemicals, enzymes, and other reagents** | | |
| β-estradiol | TCI | E0025 |
| NaCl | Merck | 1.06404 |
| Protease Inhibitor Cocktail Set III, EDTA free | Thermo Scientific | A32965 |
| IPTG | Sigma-Aldrich | I6758-5G |
| LB broth | Biobasic | SD7002 |
| MES hydrate | Sigma-Aldrich | M8250 |
| MgCl$_2$·6H$_2$O | NACALAI TESQUE | 20908 |
| Glycerol | Biobasic | C11056308 |
| Alexa Fluor 488 | Thermo Fisher Scientific | A20000 |
| Phanta DNA Polymerase | Vazyme | P515 |
| ClonExpress II One Step Cloning Kit | Vazyme | C112 |
| **Software** | | |
| Fiji | https://imagej.net/software/fiji/downloads | Fiji-win64 |
| Prism 9 | https://www.graphpad.com/features | 9.5.0 |
| Python | https://www.python.org/ | 3.8.10 |
| scikit-learn | https://scikit-learn.org/stable/ | 1.1.3 |
| XGBoost | https://github.com/dmlc/xgboost | 1.6.2 |
| UMAP | https://github.com/lmcinnes/umap | 0.5.3 |
| LocalColabFold | https://github.com/YoshitakaMo/localcolabfold | 1.5.1 |
| localCIDER | https://pappulab.github.io/localCIDER/ | 0.1.20 |
| PyMol | https://pymol.org/2/ | 2.5 |
| **Other** | | |
| HisTrap HP | GE Healthcare | 17524801 |
| HiPrep Desalting column | GE Healthcare | 17508701 |
| HiLoad Superdex 200 pg preparative SEC columns | GE Healthcare | 28989335 |

### Training and testing dataset construction

We sourced data for positive training dataset (POS) from five expansive LLPS databases: LLPSDB v2.1 (Wang et al, 2022a), PhaSePro (Mészáros et al, 2020), DrLLPS (Ning et al, 2020), PhaSepDB2.1 (Hou et al, 2023), CD-CODE (Rostam et al, 2023). In addition, we included experimentally validated phase separation protein sequences handpicked from the scientific literature. All data sources are detailed in Table EV1. Filters were set to select protein constructs that can undergo phase separation by themselves, also referred to as scaffold. Since these databases offer different formats and annotations, we manually sifted through the sequences to ensure the reliability of our positive training dataset. Only the experimentally confirmed LLPS scaffold proteins were added to the

training set. To eliminate sequences with identity exceeding 90%, we utilized CD-HIT (Fu et al, 2012) This resulted in a total of 606 unique sequences for the positive training set. For external validation, we employed a dataset from Saar et al (Saar et al, 2021). This dataset, which comprises 160 sequences recognized for their high phase separation propensity, was drawn from PhaSepDB2.1 (Hou et al, 2023).

Our negative dataset, encompassing both training and testing, came from PDB (Berman et al, 2000), a repository of structured proteins unlikely to undergo phase separation. We adopted the negative dataset of DeePhase (Saar et al, 2021), which contained entirely structured single-chain proteins. After applying a strict 30% similarity cutoff, we randomly chose 1362 sequences for the negative training dataset, while another 160 sequences comprised the negative testing set.

### Protein features extraction

A comprehensive list of feature extraction methods and associated tools can be found in Table EV2. We standardized all charge calculations to a pH of 7.4. Unless mentioned specifically in Table EV2, default parameters were used. All computational tasks were performed on the Linux Ubuntu 20.04.4 operating system (Canonical Ltd).

### Machine model building

We employed a range of machine-learning models, such as support vector machine, decision tree, Gaussian Naïve Bayes, random forest, neural network, and adaptive boosting. These were executed in scikit-learn 1.1.3 (Pedregosa et al, 2011) using default parameters. We also used the extreme gradient boosting (XGBoost) model in xgboost 1.6.2, which offers a speedy and efficient implementation of gradient-boosted decision trees (Chen and Guestrin, 2016). To prune ambiguous and redundant samples from the negative training set, we adopted one-sided selection from Python's imbalanced-learn 0.9.1 (Lemaître et al, 2017). We set n_neighbors to 1 and n_seeds_S to 45. To normalize the input variables, we applied the StandardScaler from Python's scikit-learn 1.1.3 package to the training dataset.

### Data dimension reduction

The dimension reduction technique uniform manifold approximation and projection (UMAP) based on neighbor graph was used (McInnes et al, 2018). The Python package umap-0.5.3 was employed, setting random_state to 42 and n_neighbors to 12.

### Model performance evaluation

During model training and evaluation, we adopted a train-test split ratio of 1:4, supplemented with a tenfold cross-validation. Moreover, several model performance metrics, including the area under the receiver operating characteristic curve (ROC), accuracy, F1 score, precision, and recall, were all calculated using scikit-learn 1.1.3 (Pedregosa et al, 2011).

### Protein structure prediction

Effector structures used in this study were predicted by LocalColabFold, a local version of ColabFold (Mirdita et al, 2022). Three models were run for each protein, and each model repeated with

three recycles. Models with the highest predicted local-distance difference test (pLDDT) were selected for each protein as representative. PyMol was used for visualization.

## Protein purification

Genes of HopA1, XopQ, and mRuby2 were cloned into the His-SUMO tag vector and subsequently introduced into *Escherichia coli* BL21 (DE3) Rosetta strain. The bacterial cultures were grown in Terrific Broth until they reached an $OD_{600}$ of 1. At that point, Isopropyl β-D-1-thiogalactopyranoside was added to a final concentration of 0.5 mM, and the cultures were incubated overnight at 20 °C. The cells were then lysed using a microfluidics LM20 microfluidizer. After centrifugation and filtration, the supernatant was loaded onto a 5 mL HisTrap column (GE Healthcare), operated via an ÄKTA™ FPLC system (GE Healthcare). The column was equilibrated with a binding buffer consisting of 20 mM HEPES, 500 mM NaCl, and 20 mM imidazole at pH 7.4. The target proteins were eluted using a gradient increase of the same buffer components but with 500 mM imidazole. The His-SUMO tags were then cleaved off by overnight treatment with SUMO protease at 4 °C. Following this, the sample was reloaded onto the HisTrap column to remove the cleaved His-SUMO tag and SUMO protease. Final purification was achieved using a HiLoad 16/600 Superdex 200 pg column (GE Healthcare) with a gel filtration buffer comprising 20 mM HEPES, 500 mM NaCl, and 10% glycerol at pH 7.4. HopS1 gene was cloned into pET-28a(+) vector, together with its native chaperone ShcS1 (Kabisch et al, 2005). The purification protocol was same as above without removing SUMO tag.

## Protein fluorophore labeling

HopS1, HopA1, and XopQ proteins were labeled using Alexa Fluor™ 488 (Thermo Scientific). The proteins were adjusted to a concentration of 2 mg/mL in ~500 mL of gel filtration buffer, then mixed with 0.1 M sodium bicarbonate and 10 μL of Alexa 488 dye (stock concentration of 0.5 mg/mL in DMSO). Alexa Fluor™ 488 was conjugated to the amines of protein by NHS ester. This mixture was incubated on a rotary shaker overnight at 4 °C. Any unbound dye was subsequently removed using a HiPrep™ 26/10 Desalting column (GE Healthcare) connected to an ÄKTA™ FPLC system.

## *Nicotiana benthamiana* transient expression

*Nicotiana benthamiana* (tobacco) plants were grown around 24 °C with 16-h-light and 8-h-dark cycle. Plasmids carrying the XVE::effector-mRuby2 construct were transformed into the *Agrobacterium tumefaciens* strain GV3101. The agrobacterium cells were cultured in LB medium at 30 °C overnight, then harvested by centrifugation and resuspended in a buffer containing 10 mM MES, 10 mM $MgCl_2$, and 200 μM acetosyringone. This suspension was incubated for an additional 2 h at 30 °C. Six-week-old tobacco plants were then inoculated using a needle-less syringe to inject the Agrobacterium suspension ($OD_{600} = 0.2$) into the abaxial leaf surface. After 24 h, 20 μM of β-estradiol was applied to both sides of the leaves to induce gene expression. Imaging was conducted 24 h post-induction. For plasmids carrying the 35 S::effector-mRuby2 were using the same expression strategy as XVE vectors without β-estradiol induction.

## Microscopy images acquire and fluorescence recovery after photobleaching (FRAP)

For in vitro droplet assay, samples were placed on clean cover slides and imaged using a Leica DMi8 (Leica Microsystems) equipped with an HCX PL APO 100×/1.4 oil objective, ORCA-Flash4.0 LT3 sCMOS camera (Hamamatsu), and a solid-state Spectra-X light engine (Lumencor). In vivo cell images were acquired on a Nikon Ti2 inverted spinning disc confocal (SDC) microscope (Nikon) equipped with a confocal spinning head (Yokogawa CSU-W1), a 100×1.45NA Plan-Apo objective, and a back-illuminated sCMOS camera (Orca-Fusion; Hamamatsu). Excitation light was provided by 488-nm/150 mW (Vortran) (for Alexa 488), 561-nm/100 mW (Coherent) (for mRuby2) laser combiner (iLAS system; GATACA Systems).

FRAP experiments were performed on the aforementioned Nikon SDC microscope. The laser of 488 nm for Alexa 488 and 561 nm for mRuby2 were set to 100% power for photobleaching. Timelapse images were acquired with an exposure time of 200 ms and interval of 800 ms in 1 min after photobleaching. All images were acquired using Metamorph software (Molecular Devices) and processed by Fiji.

## Kyoto Encyclopedia of Genes and Genomes (KEGG) enrichment analysis

Proteome sequences of *Xcc* 8004 and *Pst* DC3000 were downloaded from the NCBI (https://www.ncbi.nlm.nih.gov/), KEGG background files were obtained from KEGG GenomeNet (Kanehisa et al, 2002). Sequences of interest were send to eggnog-mapper (Cantalapiedra et al, 2021) to map respective KEGG pathway k-number. Fisher's exact test was conducted to calculate the significance of each pathway by using the Python package, SciPy (Virtanen et al, 2020).

# Data availability

The datasets used in this study are available in the appendix files. All materials used in this study are available by request from the corresponding author. Source data are provided with this paper.

# Peer review information

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

## Acknowledgements

This study was supported by MOE Tier 2 (MOE-T2EP30121-0015 and MOE-T2EP30122-0021 to YM), National Research Foundation Singapore NRF-NRFI08-2022-0012; NRF2021-QEP2-03-P10; OF-IRG MOH-000955 that is administered by the Singapore Ministry of Health's National Medical Research Council, and MOE Tier 3 (MOE2019-T3-1-012) to YM in Singapore.

## Author contributions

**Qiyu Liang**: Conceptualization; Data curation; Software; Formal analysis; Validation; Investigation; Methodology; Writing—original draft; Writing—review and editing. **Nana Peng**: Conceptualization; Resources; Data curation; Software; Formal analysis; Validation; Investigation; Visualization; Methodology; Writing—original draft. **Yi Xie**: Formal analysis; Methodology. **Nivedita Kumar**: Visualization; Methodology. **Weibo Gao**: Supervision; Funding acquisition; Writing—review and editing. **Yansong Miao**: Conceptualization; Formal analysis; Supervision; Funding acquisition; Validation; Investigation; Visualization; Methodology; Writing—original draft; Project administration; Writing—review and editing.

## Disclosure and competing interests statement

The authors declare no competing interests.

# Expanded View Figures

**Figure EV1.  Phase separation predictor performance.**

(A) Workflow for constructing the phase separation predictor. (B) 2D vector projection of training datasets prior to one-sided selection undersampling by UMAP. (C) Efficacy of seven clustering algorithms on a consistent training set. (D–G) Confusion matrices for external dataset predictions by (D) DeePhase, (E) PSPredictor, (F) FuzDrop, and (G) PSPer. FuzDrop's cutoff is 0.6, while the others are 0.5, as suggested by their respective studies. PSPer could not process 5 sequences for unspecified reasons, which are excluded from the matrix. Deep green indicates true outcomes, while light green indicates false outcomes. (H, I) Correlation coefficient of seven features in (H) positive training set and (I) negative training set. Source data are available online for this figure.

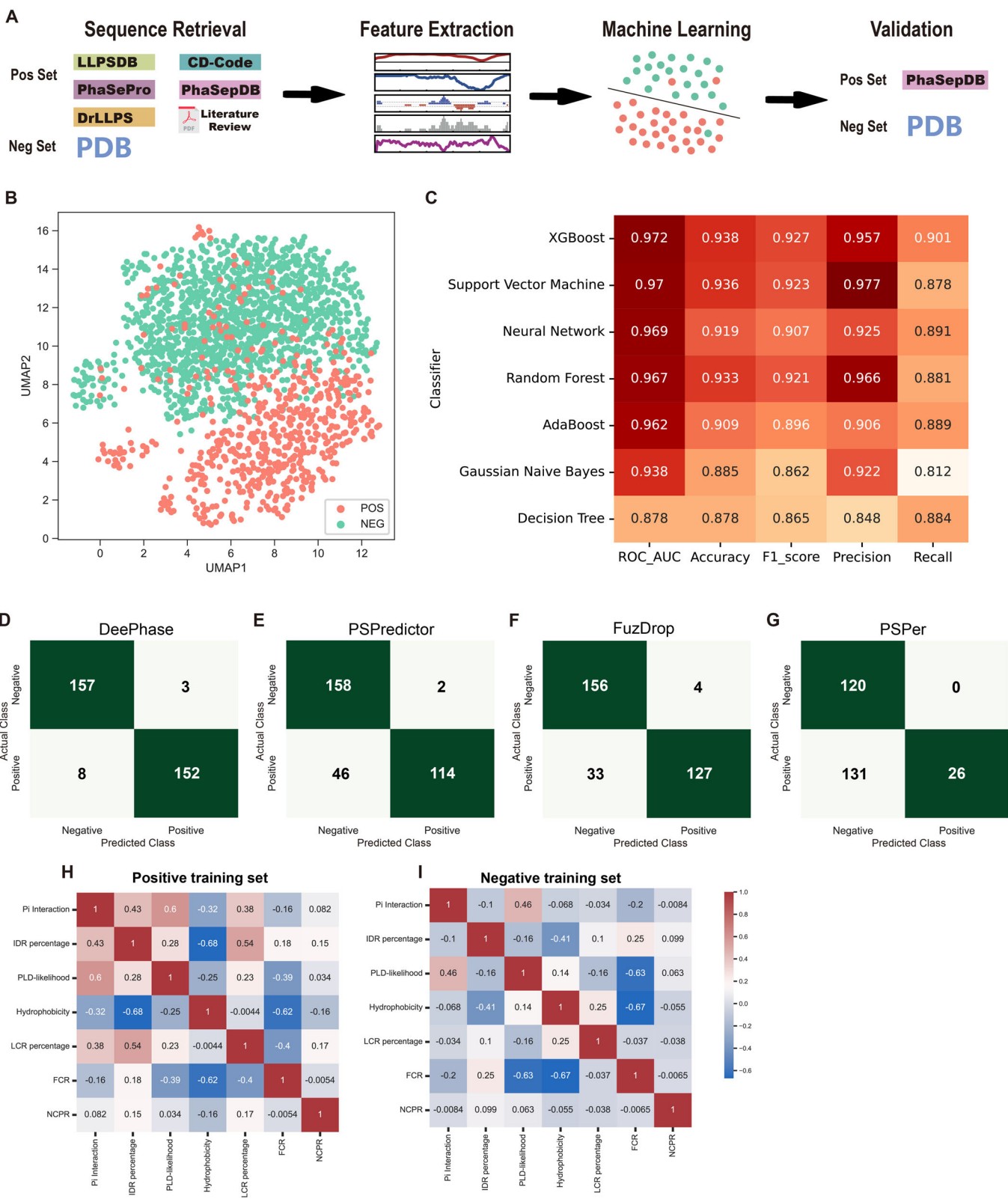

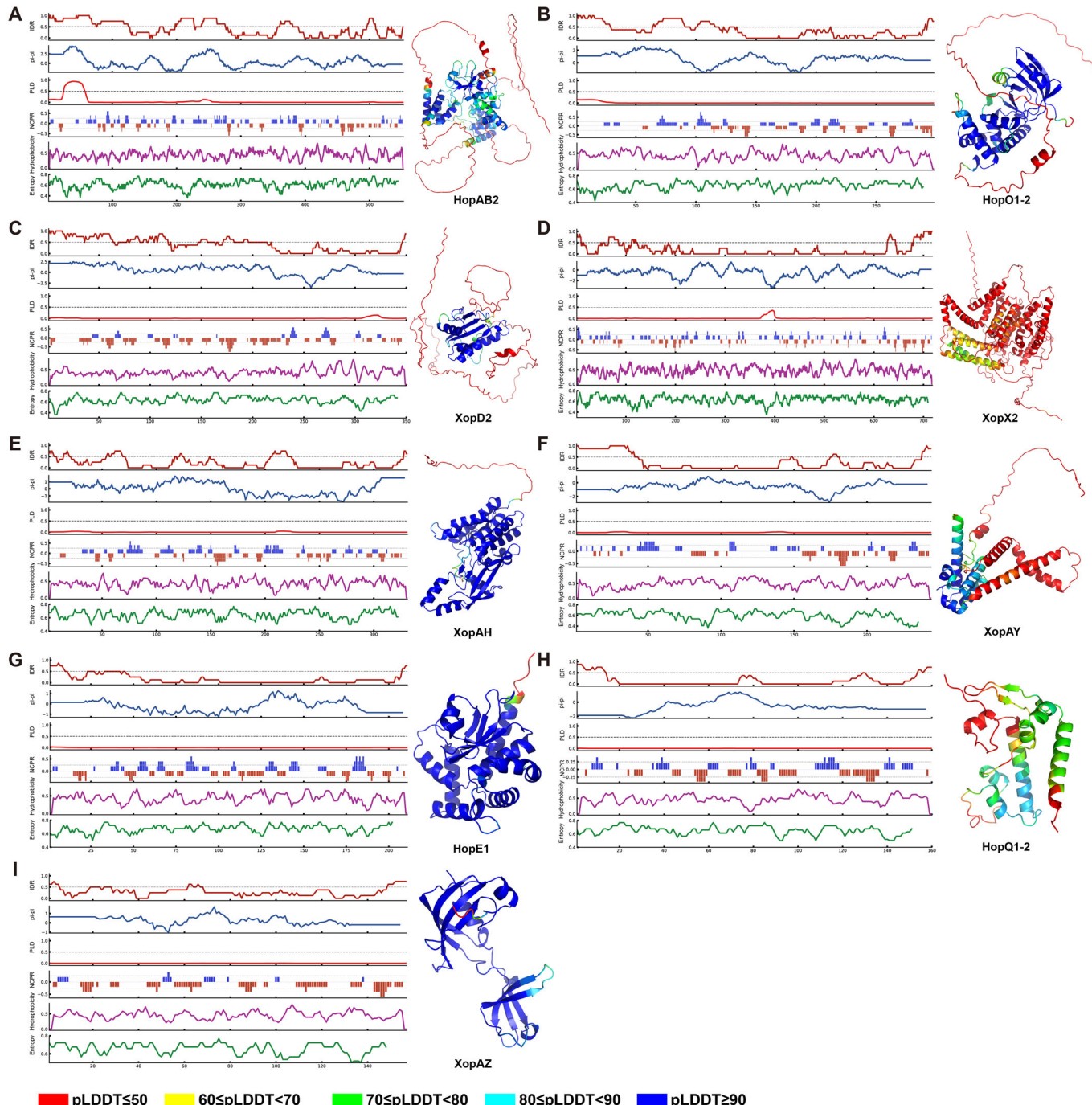

**Figure EV2. Effector protein feature extraction and structural prediction.**

(A–I) Feature extraction and AlphaFold2 structural prediction for four positive candidates: (A) HopAB2, (B) HopO1-2, (C) XopD2, (D) XopX2, (E) XopAH, (F) XopAY, (G) HopE1, (H) HopQ1-2 and (I) XopAZ. Features, from top to bottom, include IDR, pi interaction, prion-like domain likelihood, net charge per residues, hydrophobicity, and Shannon entropy. AlphaFold2-predicted structures are color-coded by predicted local-distance difference test (pLDDT) shown at the bottom of image. Structure image sizes are not to scale. This relates to Fig. 3. Source data are available online for this figure.

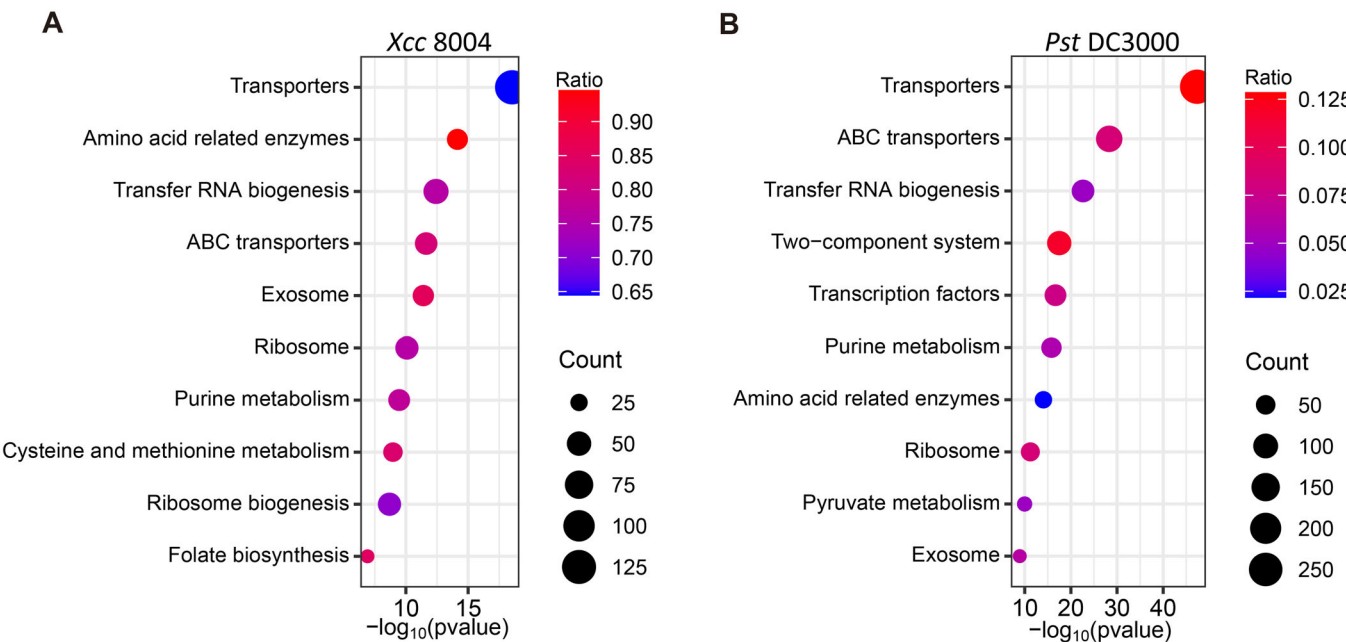

**Figure EV3. KEGG analysis for phase separation negative proteins.**

(A, B) KEGG enrichment analysis for proteins with phase separation prediction scores below 0.1 in (A) *Xcc* 8004 and (B) *Pst* DC3000. The x axis shows the −log$_{10}$ P value for the Fisher exact test, the y axis lists enrichment items, the ratio indicates the percentage of the entire pathway enriched, and the count shows the number of enriched items. Source data are available online for this figure.

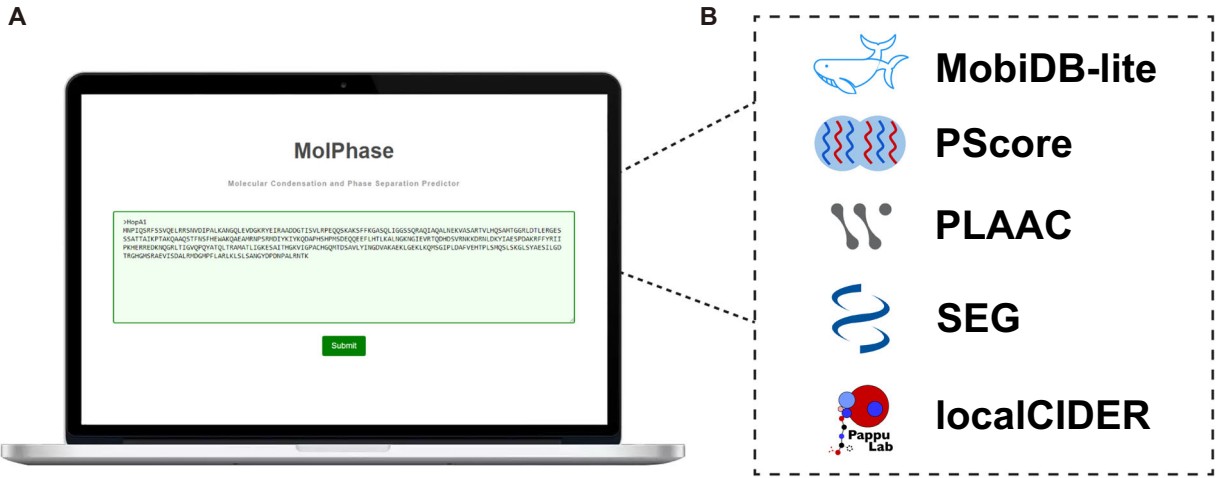

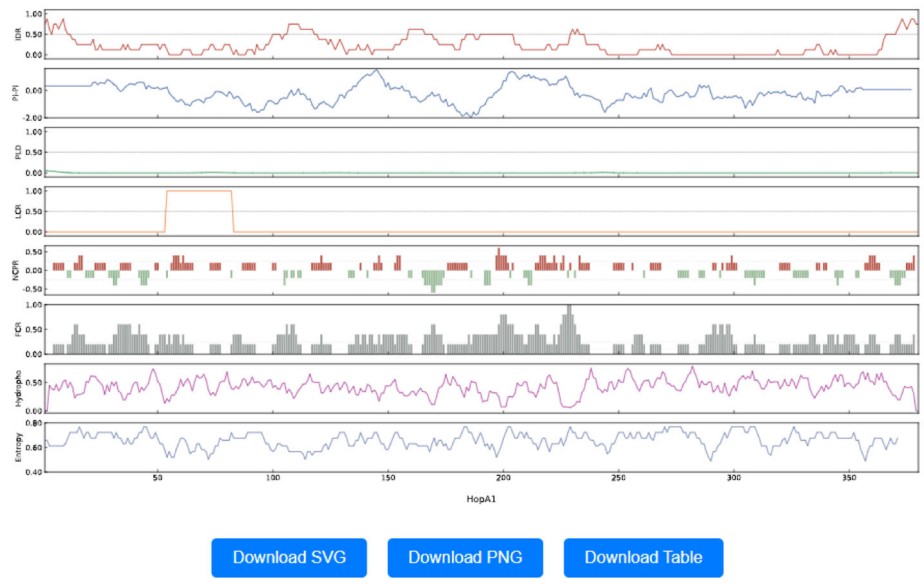

**Figure EV4.  MolPhase online predictor interface.**

(**A**) A screenshot of the MolPhase predictor, displaying the input sequence for the effector HopA1. (**B**) Tools utilized for illustrating features in (**C**). IDR was determined by Mobidb-lite, pi–pi contacts by PScore, LCR by SEG, and PLD by PLAAC. NCPR, FCR, hydrophobicity, and Shannon Entropy were assessed by localCIDER. The displayed score is the phase separation prediction score, ranging from 0 to 1. (**C**) Specific features aiding in predicting potential phase separation proteins, using HopA1 as an example. Features, from top to bottom, are IDR, pi interaction, PLD, LCR, NCPR, FCR, hydrophobicity, and Shannon Entropy. (**D**) Features illustrate in the amino acid sequence view, and feature higher than the threshold score will be highlighted.

