## [Peer Review File · The EMBO Journal]

MolPhase, an advanced prediction algorithm for protein phase separation

Qiyu Liang, Nana Peng, Yi Xie, Nivedita Kumar, Weibo Gao, and Yansong Miao

Corresponding author(s): Yansong Miao (yansongm@ntu.edu.sg)

Review Timeline:

Submission Date:	20th Sep 23
Editorial Decision:	3rd Nov 23
Revision Received:	3rd Feb 24
Editorial Decision:	21st Feb 24
Revision Received:	27th Feb 24
Accepted:	14th Mar 24

Editor: William Teale

Transaction Report:

Dear Yansong,

Thank you again for the submission of your manuscript entitled "MoIPhase: An Advanced Phase Separation Predictor and an Investigation of Phytobacterial Effector in Plant". We have now received the referees' reports, which I have copied to the bottom of this message. I would also like to apologise for the unusually long time it has taken to collect these reports.

As you can see, the reports are generally supportive of your manuscript, but express some significant concerns. All referees agree that, at its heart, the work is technically accomplished. They also state unambiguously that the manuscript is timely and the topic is important. However, the feedback was not unambiguously positive. Firstly, referee #3 asks for a wider test of MoPhase's predictive power, with more stringent negative controls. Secondly, all referees would like to see a more direct comparison between MoIPhase and other LLPS predictors before your manuscript can be published in EMBO Journal.

On balance though, I would like to invite you to address the comments of all referees in a revised version of the manuscript. I should add that it is The EMBO Journal policy to allow only a single major round of revision and that it is therefore important to resolve the main concerns at this stage. I believe the concerns of the referees are reasonable and addressable, but please contact me if you have any questions, need further input on the referee comments or if you anticipate any problems in addressing any of their points. Please, follow the instructions below when preparing your manuscript for resubmission.

I would also like to point out that as a matter of policy, competing manuscripts published during this period will not be taken into consideration in our assessment of the novelty presented by your study ("scooping" protection). We have extended this 'scooping protection policy' beyond the usual 3 month revision timeline to cover the period required for a full revision to address the essential experimental issues. Please contact me if you see a paper with related content published elsewhere to discuss the appropriate course of action.

Again, please contact me at any time during revision if you need any help or have further questions.

Thank you very much again for the opportunity to consider your work for publication. I look forward to your revision.

Best regards,

William

William Teale, Ph.D.
Editor
The EMBO Journal

When submitting your revised manuscript, please carefully review the instructions below and include the following items:

- 1) a .docx formatted version of the manuscript text (including legends for main figures, EV figures and tables). Please make sure that the changes are highlighted to be clearly visible.
- 2) individual production quality figure files as .eps, .tif, .jpg (one file per figure).
- 3) a .docx formatted letter INCLUDING the reviewers' reports and your detailed point-by-point response to their comments. As part of the EMBO Press transparent editorial process, the point-by-point response is part of the Review Process File (RPF), which will be published alongside your paper.
- 4) a complete author checklist, which you can download from our author guidelines ([https://wol-prod-cdn.literatumonline.com/pb-assets/embo-site/Author Checklist%20-%20EMBO%20J-1561436015657.xlsx](https://wol-prod-cdn.literatumonline.com/pb-assets/embo-site/Author%20Checklist%20-%20EMBO%20J-1561436015657.xlsx)). Please insert information in the checklist that is also reflected in the manuscript. The completed author checklist will also be part of the RPF.
- 5) Please note that all corresponding authors are required to supply an ORCID ID for their name upon submission of a revised manuscript.
- 6) We require a 'Data Availability' section after the Materials and Methods. Before submitting your revision, primary datasets produced in this study need to be deposited in an appropriate public database, and the accession numbers and database listed

under 'Data Availability'. Please remember to provide a reviewer password if the datasets are not yet public (see <https://www.embopress.org/page/journal/14602075/authorguide#datadeposition>). If no data deposition in external databases is needed for this paper, please then state in this section: This study includes no data deposited in external repositories. Note that the Data Availability Section is restricted to new primary data that are part of this study.

Note - All links should resolve to a page where the data can be accessed.

8) For data quantification: please specify the name of the statistical test used to generate error bars and P values, the number (n) of independent experiments (specify technical or biological replicates) underlying each data point and the test used to calculate p-values in each figure legend. The figure legends should contain a basic description of n, P and the test applied. Graphs must include a description of the bars and the error bars (s.d., s.e.m.).

9) We would also encourage you to include the source data for figure panels that show essential data. Numerical data can be provided as individual .xls or .csv files (including a tab describing the data). For 'blots' or microscopy, uncropped images should be submitted (using a zip archive or a single pdf per main figure if multiple images need to be supplied for one panel). Additional information on source data and instruction on how to label the files are available at .

10) We replaced Supplementary Information with Expanded View (EV) Figures and Tables that are collapsible/expandable online (see examples in <https://www.embopress.org/doi/10.15252/embj.201695874>). A maximum of 5 EV Figures can be typeset. EV Figures should be cited as 'Figure EV1, Figure EV2" etc. in the text and their respective legends should be included in the main text after the legends of regular figures.

12) Our journal encourages inclusion of *data citations in the reference list* to directly cite datasets that were re-used and obtained from public databases. Data citations in the article text are distinct from normal bibliographical citations and should directly link to the database records from which the data can be accessed. In the main text, data citations are formatted as follows: "Data ref: Smith et al, 2001" or "Data ref: NCBI Sequence Read Archive PRJNA342805, 2017". In the Reference list, data citations must be labeled with "[DATASET]". A data reference must provide the database name, accession number/identifiers and a resolvable link to the landing page from which the data can be accessed at the end of the reference. Further instructions are available at .

Further instructions for preparing your revised manuscript:

- a point-by-point response to the referees' comments, with a detailed description of the changes made (as a word file).
- a word file of the manuscript text.
- individual production quality figure files (one file per figure)
- a complete author checklist, which you can download from our author guidelines (<https://www.embopress.org/page/journal/14602075/authorguide>).

- Expanded View files (replacing Supplementary Information)

We realize that it is difficult to revise to a specific deadline. In the interest of protecting the conceptual advance provided by the work, we recommend a revision within 3 months (1st Feb 2024). Please discuss the revision progress ahead of this time with the editor if you require more time to complete the revisions. Use the link below to submit your revision:

Referee #1:

In the article entitled "MolPhase: An advanced phase separation predictor and an investigation of phytobacteria effector in plant," Liang, Peng, and colleagues develop a new computational predictor for phase separation and test their predictions on phytobacterial type III effectors using biochemical and cellular assays. The authors first show the power of their new prediction tool and how it compares with currently available predictors. Using MolPhase, they predict the propensity for 529 type III effectors to undergo phase separation. The authors then selected three type II effectors and mRuby2 to test their propensities for phase separation, as predicted by MolPhase. The authors found that MolPhase accurately predicted the ability of their proteins to undergo phase separation. They also found that the dynamic properties of condensates differed in their biochemical and cellular assays.

Overall, this is a well-designed study that provides the community with a more accurate tool to predict phase separation of proteins of interest. Furthermore, testing their predictor using type III effectors, proteins that are not well studied through the lens of phase separation provides proof-of-principle evidence for the successful use of MolPhase and advances knowledge of type III effector condensates. If the authors can address the concerns listed below, this reviewer will support the publication of this manuscript in the EMBO Journal.

Concerns:

- 1) In the abstract, it was unclear what the authors meant by "the phase separation of T3Es were evolved both in vivo and in vitro". This should be clarified.
- 2) How does MolPhase compare with FAIDR (PMID 33616531). The authors should comment on this in their introduction.
- 3) In line 66 of the current version of the manuscript, the authors cite phase separation prediction algorithms. They should explicitly list them in the sentence so that the reader doesn't have to sort through the literature to learn what algorithms the authors refer to.
- 4) At the end of the introduction and in the results, the authors state that the difference between biochemical and cellular dynamic properties of type III effector condensates is homo- vs. heterotypic interactions. This seems to be a limited view as a number of studies demonstrate that small molecules, salt concentrations, and pH can also alter phase separation properties. While these are discussed, it would be better to include them as possibilities wherever dynamic properties are discussed in the manuscript.
- 5) In lines 112 and 113 of the current version of the manuscript, the authors state that tightly-fold structured proteins will not undergo multivalent interaction-based phase separation. This is a bit misleading, as folded proteins, such as lysozyme, can

undergo phase separation through multiple surface-surface interactions, depending on experimental conditions. This statement should be revised to reflect this reality.

6) In line 155 of the current version of the manuscript, "directly drive PS" should be supported by citations of PMIDs 27392146 and 36603581.

7) In lines 159-160 of the current version of the manuscript, "we observed differences in the composition of amino acids between the POS and NEG datasets" should be explicitly summarized in the text so that the reader does not have to interpret Figure 1N without some guidance from the authors.

8) In line 168 of the current version of the manuscript and Fig EV1B shows that there is a trend of separation between the two groups, however there are a number of crossovers that would be incorrectly categorized based on the features selected by the authors. Would increasing the number of sequence features in the analysis improve this result and potentially reduce the number of mis-categorized proteins shown in Fig. 2C? Or is 39 sequence features the minimum number of feature that gives the most accurate predictions and additional features don't improve the predictions? Some logical explanation for why 39 features were chosen for MolPhase would be helpful as well as supplemental information showing why 39 is the chosen number. At the moment, it seems a bit arbitrary.

9) In line 228 of the current version of the manuscript, the authors state that selected proteins represented a range of MolPhase propensities. Was there a correlation between MolPhase propensities and the propensity of the protein to undergo phase separation, i.e., did a higher MolPhase propensity correlate with a lower critical concentration or some other feature of phase separation? If this was included in the text, it wasn't clear to this reviewer.

10) In lines 279-280 of the current version of this manuscript, include the percentage of the 529 type III effectors that are predicted to undergo phase separation.

11) Generally, what are the restrictions for MolPhase? Is there a minimum or maximum sequence length for confidently predicting phase separation?

12) Lines 388-389 of the current version of the manuscript should include citations for PMIDs 27056844, 3084600, and 31268421.

13) In line 492 of the current manuscript, what was the conjugation method for Alexa Fluor dye? Maleimide? NHS-ester?

14) Include additional labels in Figure 3. In A, include an arrow or some indication of the point at which MolPhase will predict phase separation on the scale to the right of the chart. For B-E, include the protein name above the figure panel so the reader can easily understand which protein is being described without having to refer to the legend. Same for Figure EV2.

15) In Figure 4, please use magenta and green rather than red and green to improve accessibility for colorblind readers.

Referee #2:

Laing et al. address a relevant need, the prediction of phase separation events in cells. The authors provide an algorithm for predicting a given protein's capacity to phase separate in cellular context. The formation of liquid droplets due to phase separation is important and timely for both cell biology and protein chemistry. Estimation of the propensity of a protein for forming liquid droplets is important for formulating a research hypothesis or interpreting data. If successful, such an algorithm will be relevant for a broad readership.,

The new predictor presented by the authors of the manuscript is more reliable and accurate than tools available so far. It analyzes a given amino acid sequence and returns the probability of phase separating as a value between 0 and 1, with 0.5 being the recommended threshold, with additional features displayed alongside amino acid sequence.

It is crucial that the authors assembled a positive training dataset of 606 experimentally validated proteins that undergo phase separation. They circumvented the challenge that selecting "tightly folded" proteins as a negative training set seems problematic. strongly hydrophobic core is required to maintain globular structure, hence compositional bias is to be expected in aa composition of such dataset. indeed, using proteins lacking disordered regions as a contrast to flexible AND undergoing phase separation shows in author's analysis in amino acid composition differences between compared sets, with phase separating proteins featuring less hydrophobic and negatively charged residues. This goes beyond the current standard set by Alberti&Hyman in the 2018 paper (Wang et al., Cell, 2018, 10.1016/j.cell.2018.06.006). The authors found that indeed, presence of aromatic residues is required for a disordered protein to phase separate.

Major points

1. The author need to address clearly whether and to which extend the algorithm presented here will go beyond further than predictors for disorder of low complexity regions.

2. Many predictors, including this one, rely on sequences from PDB. The PDB mainly catalogues well-structured proteins or domains, often omitting disordered regions that might not undergo phase separation. The authors need to demonstrate that the prediction potential goes beyond the identification of low complexity regions.

Please check the prediction quality for the DisProt database of disordered proteins.

a. How many % of these proteins are positive in the algorithm presented here?

b. Can the authors demonstrate that the prediction separate phase-separating proteins from disorder?

3. It is doubtful whether the disordered transcription factor assigned as non phase separating would indeed not form puncta at lower salt concentration or increased molarity of the recombinant protein. E.g. the positively evaluated protein HopS1 successfully undergoes phase separation at salt concentration below 25 mM or requires higher amount of protein, while negative control XopQ shows small puncta at 50 mM salt.

a. Please check the transcription factor used here for punch formation at low salt concentration.

b. Datapoints with lower salt concentration are not shown [Fig. 3 A and C]. Please add such data.

4. Only one negative control was tested among the group of transcription factors. Please add more examples to make sure the concept is not dependent on a single protein.

Referee #3:

Although it is merging that phase separation of proteins is rather sophisticated such that it's difficult to predict, still the prediction algorithms can aid the identification and characterization of phase separation in cells. There has already been a great number of algorithms based on primary sequence feature, deep learning, etc. out there used for different purposes. In this manuscript, Liang et al., established a machine-learning-based algorithm for prediction of phase separation proteins. The advantage of this algorithm is that multiple parameters were taken into account. In general, I find this work interesting but improvements have to be made.

To train the algorithm, the authors used 606 experimental-derived phase separation proteins for positive data set and 1367 well-folded proteins as negative data set. They extracted the features that can be used to define phase separation proteins by comparing the two datasets. The authors claimed that a total of 39 features were defined, which however, is exaggerated, because the amino acid composition shown in Figure 1N should be considered as one feature: relative ratio of amino acid in one protein, if I am understanding it correctly. It is not clear to me how these features are fed to the machine learning, for example, in Figure 1N, the percentage of amino acid, is it a range? if yes, the range of many residues between POS and NEG largely overlap. The authors should explain more clearly to the readers.

Also, although many features were defined, it is still based on IDR and PLD prediction and some features such as FCR, NCPR and hydrophobicity can be derived from IDR prediction. Therefore, the authors should pinpoint exactly what parameters used here had advantage over other predictors.

My concern is how much advance the present predictor can achieve than the available ones. The authors tested the efficacy of their algorithm by in parallel comparing with four other previously published predictors. Here they used the POS data set, by default, the MolPhase should behave the best because it is trained with the same dataset. Therefore, this comparison is not valid.

The authors then used MolPhase to evaluate phytobacterial effectors' phase separation in plant host. It is nice that they found HopA1, an effector predicted not to phase separate by other predictors, was predicted to have phase separation ability by MolPhase and proved to phase separation by experiment. But only one case is not sufficient, the authors can focus more on testing the phase separation of MolPhase-specific effectors such as AvrBs1, XopAH, XopAY, etc. in vitro and in vivo.

I am not sure if it's a good choice to analyze the proteomes of Xcc 8004 and Pst DC3000 as less is known about the phase separation of those bacteria. Instead, the authors can run MolPhase prediction with proteomes of Arabidopsis, for instance, since more proteins have been known to phase separate and can serve as hits for quality control.

Minor points:

In lines 112 and 113, the authors stated that homogeneously tightly-fold structured protein will not undergo multivalent interaction based PS, therefore they used 1367 proteins as negative dataset for algorithm training, it's not quite true, protein with domains that can undergo oligomerization and polymerization may also phase separate with multivalent interactions.

To me, the Kyoto Encyclopedia of Genes and Genomes (KEGG) enrichment analysis in Figures 5J and 5K are not really informative because the total number of proteins are too less.

Referee #1:

In the article entitled "MolPhase: An advanced phase separation predictor and an investigation of phytobacteria effector in plant," Liang, Peng, and colleagues develop a new computational predictor for phase separation and test their predictions on phytobacterial type III effectors using biochemical and cellular assays. The authors first show the power of their new prediction tool and how it compares with currently available predictors. Using MolPhase, they predict the propensity for 529 type III effectors to undergo phase separation. The authors then selected three type II effectors and mRuby2 to test their propensities for phase separation, as predicted by MolPhase. The authors found that MolPhase accurately predicted the ability of their proteins to undergo phase separation. They also found that the dynamic properties of condensates differed in their biochemical and cellular assays.

Overall, this is a well-designed study that provides the community with a more accurate tool to predict phase separation of proteins of interest. Furthermore, testing their predictor using type III effectors, proteins that are not well studied through the lens of phase separation provides proof-of-principle evidence for the successful use of MolPhase and advances knowledge of type III effector condensates. If the authors can address the concerns listed below, this reviewer will support the publication of this manuscript in the EMBO Journal. Concerns:

We are grateful for the positive feedback from the reviewer regarding our phase separation prediction engine. Additionally, we extend our thanks to the reviewer for providing valuable and constructive comments to enhance the manuscript. All the points raised have been thoroughly addressed.

1) In the abstract, it was unclear what the authors meant by "the phase separation of T3Es were evolved both in vivo and in vitro". This should be clarified.

We apologize for the confusion. We have rephrased the sentence now in the abstract on page 2 lines 26-29.

"The physicochemical characteristics of T3Es dictate their patterns of association for multivalent interactions, influencing the material properties of phase-separating droplets based on the surrounding microenvironment in vivo or in vitro."

2) How does MolPhase compare with FAIDR (PMID 33616531). The authors should comment on this in their introduction.

Thank you for your excellent comment. This is an important point for us to distinguish MolPhase from FAIDR regarding objectives and methodologies. MolPhase differs from FAIDR in several key aspects:

FAIDR functions as an analytical model, concentrating on gathering and comparing the biophysical features among different IDRs to potentially offer a functionally relevant classification. In contrast, MolPhase utilizes a machine learning strategy to assess the likelihood of proteins undergoing phase separation. Its focus is on understanding the molecular condensation process by analyzing how various biophysical properties can catalyze this process, aligning with different goals.

FAIDR was constructed based on the budding yeast proteome, limiting its classification to existing data and databases specific to yeast. MolPhase, on the other hand, allows real-time analysis and display of all functional features for any query proteins.

While FAIDR exclusively analyzed IDRs and provided an R-package for IDR clustering, MolPhase integrates multiple features, emphasizing that IDR is not the sole crucial feature. This is illustrated in Figure 2D and the newly added Figure EV1H, I.

Now, we described these differences between FAIDR and MolPhase on page 17 lines 448-455 .

3) In line 66 of the current version of the manuscript, the authors cite phase separation prediction algorithms. They should explicitly list them in the sentence so that the reader doesn't have to sort through the literature to learn what algorithms the authors refer to.

We appreciate the comments. We have now listed their names explicitly at the same position on page 3 lines 66-68.

4) At the end of the introduction and in the results, the authors state that the difference between biochemical and cellular dynamic properties of type III effector condensates is homo- vs. heterotypic interactions. This seems to be a limited view as a number of studies demonstrate that small molecules, salt concentrations, and pH can also alter phase separation properties. While these are discussed, it would be better to include them as possibilities wherever dynamic properties are discussed in the manuscript.

We agree with the reviewer's points and appreciate the constructive feedback. We apologize for any potential confusion caused by our previous wording. To address this, we have revised the statement 'formation of heterotypic PS in vivo by recruiting other plant components' to 'This implies that diverse microenvironments, characterized by distinct combinations of biophysical conditions, may impact the significance of intrinsic features in a particular environmental setting. Consequently, this tuning effect influences the interactions and assembly patterns of each protein undergoing phase separation.' at the introduction on page 4 lines 90-94.

Furthermore, on page 11 lines 281-285, we have better discussed how diverse micro-biophysical environments could influence inter- and intramolecular interactions, leading to the changes in the assembly of phase-separating condensates. This discussion now includes references to various factors, such as small molecules, pH, ROS, and crowding.

5) In lines 112 and 113 of the current version of the manuscript, the authors state that tightly-fold structured proteins will not undergo multivalent interaction-based phase separation. This is a bit misleading, as folded proteins, such as lysozyme, can undergo phase separation through multiple surface-surface interactions, depending on experimental conditions. This statement should be revised to reflect this reality.

Thank you for your valuable suggestions, and we fully align with the comments and perspective of the reviewer. Our initial assertions were excessively definitive, and we have accordingly rephrased them in the revised manuscript. We acknowledge that certain typically well-structured proteins, such as BSA and lysozyme, may undergo phase separation under atypical conditions, even if they do not exhibit such behavior under physiological conditions. This clarification has been incorporated into the statement on page 5 lines 115-123.

6) In line 155 of the current version of the manuscript, "directly drive PS" should be supported by citations of PMIDs 27392146 and 36603581.

Thanks for your suggestions; two citations have been added now. Currently on page 6 line 165.

7) In lines 159-160 of the current version of the manuscript, "we observed differences in the composition of amino acids between the POS and NEG datasets" should be explicitly summarized in the text so that the reader does not have to interpret Figure 1N without some guidance from the authors.

Thank you for recommending further elaboration and clarification of our statement. We have enhanced and provided more details on the features aligned with amino acids, now documented on page 7 lines 169-177. Additionally, we also emphasize that the analysis and interpretation of amino acid composition in Figure 1N could be cross-referenced with the information presented in Figure 1M.

8) In line 168 of the current version of the manuscript and Fig EV1B shows that there is a trend of separation between the two groups, however there are a number of crossovers that would be incorrectly categorized based on the features selected by the authors. Would increasing the number of sequence features in the analysis improve this result and potentially reduce the number of mis-categorized proteins shown in Fig. 2C? Or is 39 sequence features the minimum number of feature that gives the most accurate predictions and additional features don't improve the predictions? Some logical explanation for why 39 features were chosen for MolPhase would be helpful as well as supplemental information showing why 39 is the chosen number. At the moment, it seems a bit arbitrary.

Thank you for the comments. Firstly, we would like to clarify that we have a total of 39 features, representing points distributed in a 39-dimensional space. To visualize these points in a 2D plane, we utilized UMAP to reduce dimensions from 39 to 2, resulting in Fig EV1B and Fig 2A as projections from the 39-dimensional space. The cross observed in 2 dimensions may represent separation in the 39-dimensional space. The machine learning model aims to create a super-surface in this 39-dimensional space for distinguishing positive and negative groups. Fig 2C shows that only 7 dots are misclassified, meaning there are only 7 crossovers in the 39-dimensional space. This clarification has been added to page 8 lines 205-206.

Secondly, the increase in the number of features indeed enhances model performance. As indicated in the table below, MolPhase, incorporating the most abundant features, achieves the best performance. The addition of features correlates with improved performance, and MolPhase integrates a comprehensive set of 39 intrinsic biophysical and chemical features related to phase separation. These features encompass all known phase separation-related intrinsic factors, providing a holistic approach. This explanation has been included on page 7 lines 180-181.

Name	Feature Number	Prediction methods	Performance claimed by themselves	True positive rate (Fig EV1D-G)	True negative rate (Fig EV1D-G)
DeePhase	5	Feature engineering combined with language model	AUC 0.83	95%	98.13%
FuzDrop	-	Logistic model	AUC 0.922	79%	97.50%
PSPer	-	Hidden Markov Model	-	16.56%	100%
PhasePred	10	XGBoost classifier	AUC 0.85	-	-
PSPredictor	-	word2vec	Accuracy 0.95±0.03	71.25%	98.75%
MolPhase	39	XGBoost classifier	AUC 0.968	98.75%	96.88%

Table R1. Comparison of different machine learning based phase separation predictors.

9) In line 228 of the current version of the manuscript, the authors state that selected proteins represented a range of MolPhase propensities. Was there a correlation between MolPhase propensities and the propensity of the protein to undergo phase separation, i.e., did a higher MolPhase propensity correlate with a lower critical concentration or some other feature of phase separation? If this was included in the text, it wasn't clear to this reviewer.

We appreciate the excellent points brought by the Reviewer. MolPhase generates prediction scores that correlate with the propensity for phase separation. However, it's crucial to emphasize that this correlation holds over a broad spectrum but is not directly comparable within a narrow range. The reasons are the following. The MolPhase score reflects a protein's intrinsic properties related to phase separation, where a higher score indicates a greater contribution to phase separation factors. Nevertheless, these contribution factors are conditionally dependent in wet lab experiments, and variations in environmental cues can lead to diverse phase behaviors.

The MolPhase score could be relatively insensitive within a narrow evaluation window for critical concentration. For instance, a protein with a score of 0.9 may undergo phase separation more quickly than one with a score of 0.5, indicating a lower critical concentration. However, a protein with a score of 0.95 doesn't necessarily show significantly lower critical concentration than a protein with a score of 0.9. The nuanced differences between 0.9 and 0.95, observable during the initial phase nucleation stage, might not be captured using an imaging approach that lacks the spatiotemporal resolution to investigate assembly status at the nucleation stage.

In short, while fundamental biophysical features are comparable, diverse experimental conditions and investigation approaches can influence the degree of differences reflected by MolPhase scores. We have now added a brief discussion on page 16 lines 428-430 to bring caution and sufficient consideration of experimental conditions when directly comparing proteins with close MolPhase scores.

Due to the complexity of microenvironments in vivo, understanding the interplay between diverse factors and protein biophysical features is essential to understanding phase separation in a broad perspective. Our ongoing work integrates protein levels, interactions, and modifications to understand what makes a functional phase separation under different conditions. We anticipate reporting and addressing this thought-provoking question from the reviewer soon.

10) In lines 279-280 of the current version of this manuscript, include the percentage of the 529 type III effectors that are predicted to undergo phase separation.

In total, 63% (335 out of 529) of PsyTEC effectors are predicted to be MolPhase score higher than 0.5, which might undergo phase separation under specific conditions. This statement also added in manuscript on page 12 lines 308-309.

11) Generally, what are the restrictions for MolPhase? Is there a minimum or maximum sequence length for confidently predicting phase separation?

MolPhase has two restrictions:

1. Only unambiguous amino acids are permitted; characters beyond the twenty basic amino acids, such as 'X', are invalid.
2. While the algorithm has no maximum sequence length restriction, the online website allows a maximum input length of 3000 amino acids due to computational resource considerations. The minimum length for prion-like domain prediction is 40 amino acids.

These restrictions are detailed on page 13 lines 340-341 of our manuscript

12) Lines 388-389 of the current version of the manuscript should include citations for PMIDs 27056844, 3084600, and 31268421.

Three citations have been added. The citations on page 17 lines 460-462 of revised manuscript. For 3084600, we believe that it should be 30846600.

30846600: A molecular assembly phase transition and kinetic proofreading modulate Ras activation by SOS, Science, 2019

3084600: Chemotherapy of East Coast fever: the long term weight changes, carrier state and disease manifestations of parvaquone treated cattle, Journal of Comparative Pathology, 1986

13) In line 492 of the current manuscript, what was the conjugation method for Alexa Fluor dye? Maleimide? NHS-ester?

The Alexa Fluor dye was conjugated through NHS-ester. Details are added now in the method on page 23 line 575.

14) Include additional labels in Figure 3. In A, include an arrow or some indication of the point at which MolPhase will predict phase separation on the scale to the right of the chart. For B-E, include the protein name above the figure panel so the reader can easily understand which protein is being described without having to refer to the legend. Same for Figure EV2.

We have now highlighted the area where MolPhase scores higher than 0.5 and changed the legends for heatmap. Additionally, we have adjusted the labels for Fig 3B-E and Fig EV2 as per reviewer's suggestions.

15) In Figure 4, please use magenta and green rather than red and green to improve accessibility for colorblind readers.

Changed according to the suggestions.

Referee #2:

Laing et al. address a relevant need, the prediction of phase separation events in cells. The authors provide an algorithm for predicting a given protein's capacity to phase separate in cellular context. The formation of liquid droplets due to phase separation is important and timely for both cell biology and protein chemistry. Estimation of the propensity of a protein for forming liquid droplets is important for formulating a research hypothesis or interpreting data. If successful, such an algorithm will be relevant for abroad readership.,

The new predictor presented by the authors of the manuscript is more reliable and accurate than tools available so far. It analyzes a given amino acid sequence and returns the probability of phase separating as a value between 0 and 1, with 0.5 being there commended threshold, with additional features displayed alongside amino acid sequence.

It is crucial that the authors assembled a positive training dataset of 606 experimentally validated proteins that undergo phase separation. They circumvented the challenge that selecting "tightly folded" proteins as a negative training set seems problematic. Strongly hydrophobic core is required to maintain globular structure, hence compositional bias is to be expected in aa composition of such dataset. indeed, using proteins lacking disordered regions as a contrast to flexible AND undergoing phase separation shows in author's analysis in amino acid composition differences between compared sets, with phase separating proteins featuring less hydrophobic and negatively charged residues. This goes beyond the current standard set by Alberti & Hyman in the 2018 paper (Wang et al., Cell, 2018, 10.1016/j.cell.2018.06.006). The authors found that indeed, presence of aromatic residues is required for a disordered protein to phase separate.

Major points

1. The author need to address clearly whether and to which extend the algorithm presented here will go beyond further than predictors for disorder of low complexity regions.

Our approach extends the evaluation of phase separation proteins beyond IDR and LCR analysis, incorporating 39 features for model construction, including pi interaction, prion-like domain, charge block cluster, etc. The feature importance profile (Fig 2D) highlights pi interaction as the most crucial feature, surpassing IDR percentage. Figure R1 demonstrates through feature correlation analysis that other features are not consistently correlated with IDR or LCR, emphasizing the need for integrating multiple features. Figure R1 is now also added as new Figure EV1H,I.

Furthermore, in assessing our model's accuracy for predicting phase separation proteins with a low proportion of IDR, we chose HopA1 as a validation example. In contrast to HopS1, which has 80.5% IDR, HopA1 only contains 8.9% IDR (page 9 lines 249-250) and features a well-folded C-terminal structure predicted by AlphaFold2. Both in vivo and in vitro experimental results for HopA1 demonstrate MolPhase's effective performance on proteins with low IDR percentages. Notably, due to HopA1's low IDR percentage, it was predicted as phase separation negative by PSpredictor, PSpPer, and FuzDrop, receiving a score of 0.51 from DeePhase. This comparison underscores MolPhase's ability to predict phase separation beyond disorder. To explore the relationship between MolPhase and disorder predictors, additional discussion has been included on page 15 lines 394-406.

Figure R1: Correlation coefficient of seven feature in (A) positive training set and (B) negative training set.

2. Many predictors, including this one, rely on sequences from PDB. The PDB mainly catalogues well-structured proteins or domains, often omitting disordered regions that might not undergo phase separation. The authors need to demonstrate that the prediction potential goes beyond the identification of low complexity regions.

We thank reviewer's comments. While we acknowledge the importance of IDR and LCR as key features of phase separation proteins, it's essential to note that IDR alone has been known does not guarantee phase separation. As highlighted in Borchers W et al., 2021 (<https://doi.org/10.1016/j.sbi.2020.09.004>), "low sequence complexity is neither required nor sufficient for an IDR to undergo phase separation." The disordered region itself does not act as the driving force for phase separation; rather, it provides flexibility. Multivalent interactions, such as hydrophobicity, charge blocking, pi-pi interaction, and prion-like domains within disordered regions, contribute to phase separation. To comprehensively capture these interaction forces, we integrated multiple features into our model. The feature contribution percentage of the final model reveals that pi interaction holds the highest importance, followed by IDR percentage. This reaffirms that MolPhase does not solely rely on disordered regions to predict phase separation. We also added discussion on page 15 lines 407-414 to discuss why PDB served as the best choice for negative dataset at this time point.

Please check the prediction quality for the DisProt database of disordered proteins.

a. How many % of these proteins are positive in the algorithm presented here?

To analyse DisProt database stored sequences, we first obtained 2629 none-repeat individual protein sequences from DisProt and then predicted their phase separation probability by MolPhase, and score distribution was plotted below. As shown in the Figure R2, protein with extremely low IDR percentage also can obtain a high MolPhase score (lower right corner), while protein with long IDR also could be predicted as phase separation negative (upper left corner), consistent with the explanation and evidence of the above comment. In total, 60.8% of protein (1599 out of 2629) were predicted to be phase separation positive. And some protein could still predict to be phase separation negative even with nearly 100% of IDR. These results together suggestion that as a multi-feature-based predictor, our machine-learning based MolPhase is able to overcome single IDR

feature based evaluation and capture the phase separation protein with comprehensive analysis and prediction.

Figure R2. Distribution of MolPhase scores based on varying IDR percentages in 2629 proteins extracted from DisProt

b. Can the authors demonstrate that the prediction separate phase-separating proteins from disorder?

The Figure R2 above offers evidence to distinguish between the IDR feature and the combined evaluation of phase separation through multi-biophysical features. Despite the commonly perceived association of features such as pi-pi interaction, hydrophobicity, FCR, etc., with IDR, our newly added feature correlation analysis in Figure R1 (new Figure EV 1H and 1I) demonstrates that their correlation coefficients are not actually relevant. This distinction between the multi-feature predictor and IDR-only analysis is now added in the discussion on page 14-15 lines 394-406.

3. It is doubtful whether the disordered transcription factor assigned as non phase separating would indeed not form puncta at lower salt concentration or increased molarity of the recombinant protein. E.g. the positively evaluated protein HopS1 successfully undergoes phase separation at salt concentration below 25 mM or requires higher amount of protein, while negative control XopQ shows small puncta at 50 mM salt.

a. Please check the transcription factor used here for punch formation at low salt concentration.

We guess the reviewer refers to “effector”, instead of “Transcription factor”, since we did not use transcription factor in the manuscript.

Thanks for the question. Here, we provide additional details on distinguishing microscopic imaging signals and differentiating between punctated oligomers and phase separating. Considering the protein assembly status, the mentioned imaging patterns can be categorized into three assembly modes: colloidal packing mediated oligomerization, percolation clustering, and phase separation.

First, colloidal packing-mediated oligomerization involves the assembly of colloidal particles into small, well-defined oligomers primarily driven by a lock-and-key binding mode dominant in well-folded domains. This results in discrete, organized oligomers rather than large-scale clustering or phase separation. When observed under a normal fluorescent microscope, as seen in the case of XopQ, it looks like punctate patterns. Alternatively, it manifests as discrete dots with clear size and intensity distribution via high-resolution single molecular imaging.

Second, moving on to percolation phenomena entails forming interconnected percolation clusters, potentially relying on flexible binding surfaces, such as multivalent short-linear motifs in IDR. This leads to the emergence of a continuous percolation network, displaying punctate patterns at low concentrations that expand to a network at higher concentrations. This is not the observed behavior for XopQ.

Third, phase separation refers to the segregation of components due to thermodynamic immiscibility or differences in affinity between components, resulting in the spontaneous separation into two or more phases. After the nucleation stage, after passing critical saturation concentration, which may resemble punctate patterns under a microscope, obvious coalescence or coarsening occurs to drive phase growth. It's important to highlight that this behavior is not observed in the case of XopQ.

In summary, each of these processes exhibits unique characteristics driven by specific mechanisms and interactions. We have also recently written a review to describe different biomolecular assembly statuses (<https://doi.org/10.1016/j.pbi.2023.102374>). Here, additional brief clarification about XopQ has been now added on page 10 lines 274-276.

b. Datapoints with lower salt concentration are not shown [Fig. 3 A and C]. Please add such data.

We through the reviewer refers to Figure 4 C and D here. We have re-done the whole set of droplet assay of XopQ and mRuby2 and now added low salt concentration for XopQ and mRuby2. Both of them still show no phase separation under 25mM NaCl.

4. Only one negative control was tested among the group of transcription factors. Please add more examples to make sure the concept is not dependent on a single protein.

Now, we have added additional three MolPhase-predicted negative effectors: HopE1, HopQ1-2 and XopAZ as shown below. All of them showed diffusive patterns *in vivo*. We added the new data in Figure 4H.

Figure R3. In vivo expression of HopE1, HopQ1-2 and XopAZ. Scale bars are 10 μm for broader images and 1 μm for close-ups.

Referee #3:

Although it is merging that phase separation of proteins is rather sophisticated such that it's difficult to predict, still the prediction algorithms can aid the identification and characterization of phase separation in cells. There has already been a great number of algorithms based on primary sequence feature, deep learning, etc. out there used for different purposes. In this manuscript, Liang et al., established a machine-learning-based algorithm for prediction of phase separation proteins. The advantage of this algorithm is that multiple parameters were taken into account. In general, I find this work interesting but improvements have to be made.

1) To train the algorithm, the authors used 606 experimental-derived phase separation proteins for positive data set and 1367 well-folded proteins as negative data set. They extracted the features that can be used to define phase separation proteins by comparing the two datasets. The authors claimed that a total of 39 features were defined, which however, is exaggerated, because the amino acid composition shown in Figure 1N should be considered as one feature: relative ratio of amino acid in one protein, if I am understanding it correctly. It is not clear to me how these features are fed to the machine learning, for example, in Figure 1N, the percentage of amino acid, is it a range? if yes, the range of many residues between POS and NEG largely overlap. The authors should explain more clearly to the readers.

We apologize for the confusion derived from our previous unclear descriptions. In Figure 1N, each amino acid's percentage within the full-length protein sequence is treated as an individual feature. This evaluation is repeated twenty times, considering the twenty amino acids as separate features rather than a single feature. This approach is taken because a single value cannot adequately represent the composition of all 20 amino acids.

Here, we aim to clarify the point without introducing technical jargon from machine learning to avoid potential confusion. Regarding amino acid composition, a single protein has a fixed value for the percentage of each amino acid, while a whole set of training proteins exhibits a distribution within a range. Due to the inherent variability among different proteins, an overlapping range of amino acid composition and distribution between POS and NEG is normal. Our machine-learning model learns various feature combinations from protein sequences. By combining the entire training set, the model can discern patterns that distinguish positive and negative instances. The model essentially identifies a boundary between the combinations of features associated with phase separation-positive and negative proteins. Therefore, although amino acid percentage distribution may overlap within a certain range, the machine learning algorithm leverages feature combinations to effectively classify positive and negative sets.

Here, we have clarified how the amino acid features were isolated and used as training features on page 7 lines 169-177.

2) Also, although many features were defined, it is still based on IDR and PLD prediction and some features such as FCR, NCPR and hydrophobicity can be derived from IDR prediction. Therefore, the authors should pinpoint exactly what parameters used here had advantage over other predictors.

Our approach extends the evaluation of phase separation proteins beyond IDR and PLD analysis, incorporating 39 features for model construction, including pi interaction, prion-like domain, charge block cluster, etc. The feature importance profile (Fig 2D) highlights pi interaction as the most crucial feature, surpassing IDR percentage. Figure R1 demonstrates through feature correlation analysis that other features are not consistently correlated with IDR or PLD, emphasizing the need for integrating multiple features. Figure R1 is now also added as new Figure EV1H,I.

Furthermore, in assessing our model's accuracy for predicting phase separation proteins with a low proportion of IDR, we chose HopA1 as a validation example. In contrast to HopS1, which has 80.5% IDR, HopA1 only contains 8.9% IDR (page 9 lines 249-250) and features a well-folded C-terminal structure predicted by AlphaFold2. Both in vivo and in vitro experimental results for HopA1 demonstrate MolPhase's effective performance on proteins with low IDR percentages. Notably, due to HopA1's low IDR percentage, it was predicted as phase separation negative by PSPredictor, PSPer, and FuzDrop, receiving a score of 0.51 from DeepPhase. This comparison underscores MolPhase's ability to predict phase separation beyond disorder. To pinpoint the relationship between MolPhase and disorder predictors as reviewer suggested, we have now added additional discussion on page 14 lines 368-371 and page 15 lines 394-406.

Figure R1: Correlation coefficient of seven feature in (A) positive training set and (B) negative training set.

In terms of the comparison with other predictors, we integrated more features than others, which mean we cover more possible interaction forces for phase separation and therefore we have better performance. Feature number comparison is shown in the below table, also shown in Figure 2E and F, and described on pages 8-9 lines 214-225.

Name	Feature Number	Prediction methods	Performance claimed by themselves	True positive rate (Fig EV1D-G)	True negative rate (Fig EV1D-G)
DeePhase	5	Feature engineering combined with language model	AUC 0.83	95%	98.13%
FuzDrop	-	Logistic model	AUC 0.922	79%	97.50%
PSPer	-	Hidden Markov Model	-	16.56%	100%
PhasePred	10	XGBoost classifier	AUC 0.85	-	-
PSPredictor	-	word2vec	Accuracy 0.95±0.03	71.25%	98.75%
MolPhase	39	XGBoost classifier	AUC 0.968	98.75%	96.88%

Table R2. Comparison of different machine learning based phase separation predictors.

3) My concern is how much advance the present predictor can achieve than the available ones. The authors tested the efficacy of their algorithm by in parallel comparing with four other previously published predictors. Here they used the POS data set, by default, the MolPhase should behave the best because it is trained with the same dataset. Therefore, this comparison is not valid.

Thanks for your comments. As depicted in Table R2, MolPhase exhibits the highest true positive rate along with a substantial true negative rate. Additionally, MolPhase assigns the highest weightage to positive proteins and the lowest weightage to negative proteins (Figure 2E, F). These collective findings indicate that MolPhase outperforms other predictors.

Here we want to clarify that the training set and testing set used in this manuscript are independent, which is not “the same dataset”. We took measures such as applying CD-HIT to ensure that the testing set did not contain sequences with 90% similarity or higher compared to the training set. This ensures the validity of our testing and reflects the genuine performance of the model.

4) The authors then used MolPhase to evaluate phyto-bacterial effectors' phase separation in plant host. It is nice that they found HopA1, an effector predicted not to phase separate by other predictors, was predicted to have phase separation ability by MolPhase and proved to phase separate by experiment. But only one case is not sufficient, the authors can focus more on testing the phase separation of MolPhase-specific effectors such as AvrBs1, XopAH, XopAY, etc. in vitro and in vivo.

We appreciate your insightful comments that have strengthened the validation of MolPhase. In response, we designed new constructs and have included in vivo tobacco expression data for XopAH and XopAY, demonstrating their distinct condensation inside plant cells (Figure R4 and Figure 4H). This addition is highlighted on page 11 lines 288-291, supporting high performance MolPhase.

Unfortunately, despite multiple attempts to optimize the in vitro protein purification conditions for more than two months, we encountered challenges in obtaining soluble XopAH and XopAY proteins for conducting in vitro phase separation assays, implying a high propensity in inter- and intramolecular interactions, which can not be solved at this moment. XopAH forms insoluble precipitation after SUMO tag cleavage, even with surfactants like Triton X-100. Additionally, XopAY resulted in an extremely low yield of purification, preventing us from reaching the required concentration for in vitro droplet assays. The challenges encountered in in vitro biochemistry and the observation of phase separation phenomena in vivo collectively reinforce the positive validation of MolPhase. The inclusion of additional in vivo data has significantly enriched our dataset. We appreciate the reviewer's wonderful suggestions.

Figure R4. In vivo expression of XopAH and XopAY. Scale bars are 10 μm for broader images and 1 μm for close-ups.

5) I am not sure if it's a good choice to analyze the proteomes of Xcc 8004 and Pst DC3000 as less is known about the phase separation of those bacteria. Instead, the authors can run MolPhase prediction with proteomes of Arabidopsis, for instance, since more proteins have been known to phase separate and can serve as hits for quality control.

We appreciate the insightful comments from the reviewer. We echo with the reviewer in a similar way in looking for a deeper and broader understanding of phase separation in life science.

Our current manuscript focuses on constructing a phase separation predictor. Actually, we are concurrently working on a separate research project from a systems biology perspective, with the manuscript currently in preparation. MolPhase predictions have been applied to analyze proteomes of more than 1000 species, employing a systems biology analytical approach to identify phase-separated protein-driven functional condensate hubs across diverse biological processes.

In response to the reviewer's comment, we present an example of Arabidopsis analysis in the response letter (Figure R5), a major model species that is extensively examined in another manuscript currently in preparation. To maintain the continuity of the narrative for the upcoming story, we have chosen not to include this Arabidopsis analysis in this manuscript. The development of phase separation predictors stands as an independent contribution. We believe the current demonstration utilizes effectors from Pst DC3000 and Xcc 8004, with an additional 529 effectors from 494 strains sourced from the Pseudomonas syringae Type III Effector Compendium, providing a substantial set for exploration and demonstration.

Minor points:

6) In lines 112 and 113, the authors stated that homogeneously tightly-fold structured protein will not undergo multivalent interaction based PS, therefore they used 1367 proteins as negative dataset for algorithm training, it's not quite true, protein with domains that can undergo oligomerization and polymerization may also phase separate with multivalent interactions.

Thank you for your clarification. We apologize for any confusion caused by the inaccurate description. We fully concur with the reviewer's perspective. In our revised manuscript, we elaborate on the reasons why the PDB database is chosen as the best available negative training data for phase separation, emphasizing the stable intra-molecular interactions and weak inter-molecular associations typical of proteins/domains with high-resolution structures. To exercise caution, we have included specific examples of well-folded proteins that undergo phase separation with multivalent weak interactions under certain experimental conditions. This additional discussion can be found on page 5 lines 115-123 and page 15 lines 407-414 of the revised manuscript.

7) To me, the Kyoto Encyclopedia of Genes and Genomes (KEGG) enrichment analysis in Figures 5J and 5K are not really informative because the total number of proteins are too less.

The limited number of KEGG enrichment results is due to the low ratio of phase separation positive proteins and the sparse annotation information for Xcc 8004 and Pst DC3000. Despite this, the KEGG enrichment analysis indicates several high correlations between potential phase separation regulation and the enriched pathway via enriched proteins. In contrast to GO terms, which have a mixed hierarchy and may offer less information in this bacteria analysis, KEGG enrichment could be a more effective classification method for predicting involved biochemical reaction-specific pathways. This information can be valuable for researchers focusing on these specific pathways.

To illustrate the enriched functional pathway classification through KEGG enrichment, we present an additional example using an Arabidopsis proteome analysis in our response letter. Similar to the enrichment results for Xcc and Pst, the KEGG enrichment of proteins with MolPhase scores higher than 0.9 in Arabidopsis highlights a richness in nucleic acid-related pathways. This underscores the significance of regulating nucleic acid-related pathways through the phase separation mechanism. Thus, we would like to keep our KEGG analysis to be shared with readers with related research interests.

Dear Yansong,

We have now received re-review reports from two referees. As you will see, you have addressed their concerns satisfactorily. Before I can finally accept the manuscript though, there are some remaining editorial points which need to be addressed. In this regard would you please:

- rename the Conflict of Interests statement to "Disclosure and Competing Interests Statement",
- remove the AC/CRedit section from the manuscript,
- include figure callouts for Fig. 4F and 4I,
- rename your dataset 'Dataset EV1', using the appropriate callout in the manuscript text and legend, and including the legend in the Excel file as a separate tab,
- similarly, rename Appendix Table S1-S2 as Table EV1-EV2 with the appropriate callouts,
- reorganize Source Data files so that file/folder contains one figure and ZIPing for each main figure; for EV figures, ZIP together all source data,
- summarise general information in a 'Data Information' section in the legends of figures 1a-n, 4g, j; 5d-l,
- label the orphan panel '4e' in figure and legend,
- rectify the mismatch between the annotated p values in the figure legend of figures 5d-i and the annotated p values in the figure file,
- define n in the legends of figures 5d-l,
- include a scale bar is missing for figures 4b-d, and
- remove Appendix files legends from ms file.

We include a synopsis of the paper (see <http://emboj.embopress.org/>). Please provide me with a two-sentence general summary statement and 3-5 bullet points that capture the key findings of the paper.

We also need a summary figure for the synopsis. The size should be 550 wide by [200-400] high (pixels). You can also use something from the figures if that is easier.

EMBO Press is an editorially independent publishing platform for the development of EMBO scientific publications.

Best wishes,

William

William Teale, PhD
Editor
The EMBO Journal
w.teale@embojournal.org

Please remember: Digital image enhancement is acceptable practice, as long as it accurately represents the original data and

conforms to community standards. If a figure has been subjected to significant electronic manipulation, this must be noted in the figure legend or in the 'Materials and Methods' section. The editors reserve the right to request original versions of figures and the original images that were used to assemble the figure.

We realize that it is difficult to revise to a specific deadline. In the interest of protecting the conceptual advance provided by the work, we recommend a revision within 3 months (21st May 2024). Please discuss the revision progress ahead of this time with the editor if you require more time to complete the revisions. Use the link below to submit your revision:

Referee #1:

The authors have sufficiently addressed each point raised by this reviewer. The clarifications added to their manuscript have greatly improved understanding during the first read. This reviewer appreciates the attention to detail for suggested citations as well. As such, this reviewer supports the publication of this manuscript in the EMBO Journal.

Referee #3:

The authors have addressed my concerns.

All editorial and formatting issues were resolved by the authors.

Dear Yansong,

I am pleased to inform you that your manuscript has been accepted for publication in the EMBO Journal.

This will, I am sure, be a really useful community resource. Congratulations!

Best wishes,

William

William Teale, PhD
Editor
The EMBO Journal
w.teale@embojournal.org
